



# Seasonal variability of the Atlantic Meridional Overturning Circulation at 11°S inferred from bottom pressure measurements

Josefine Herrford[1], Peter Brandt[1,2], Torsten Kanzow[3], Rebecca Hummels[1], Moacyr Araujo[4], Jonathan V. Durgadoo[1,2]

[1]GEOMAR Helmholtz Centre for Ocean Research, Kiel, Germany
[2]Christian-Albrechts-Universität zu Kiel, Kiel, Germany
[3]Alfred Wegener Institute, Bremerhaven, Germany
[4]Department of Oceanography Federal University of Pernambuco, Recife, Brazil

*Correspondence to*: Josefine Herrford (jherrford@geomar.de)

**Abstract.** Bottom pressure observations on both sides of the Atlantic basin, combined with satellite measurements of sea level anomalies and wind stress data, are utilized to estimate variations of the Atlantic Meridional Overturning Circulation (AMOC) at 11° S. Over the period 2013-2018, the AMOC and its components are dominated by seasonal variability, with peak-to-peak

amplitudes of 12 Sv for the upper-ocean geostrophic transport, 7 Sv for the Ekman and 14 Sv for the AMOC transport. The observed seasonal cycles of the AMOC, its components as well as the Western Boundary Current as observed with current meter moorings are in general good agreement with results of an ocean general circulation model. The seasonal variability of zonally integrated geostrophic velocity in the upper 300 m is controlled by pressure variations at the eastern boundary, while at 500 m depth contributions from the western and eastern boundaries are similar. The model tends to underestimate the

seasonal pressure variability at 300 and 500 m depth, slightly stronger at the western boundary. In the model, seasonal AMOC variability at 11° S is governed by the variability in the eastern basin. Here, long Rossby waves originating from equatorial forcing are known to be radiated from the Angolan continental slope and propagate westward into the basin interior. The contribution of the western basin to AMOC seasonal variability is instead comparably weak as transport variability due to locally forced Rossby waves is mainly compensated by the Western Boundary Current. Our analyses indicate, that while some

of the uncertainties of our estimates result from the technical aspects of the observational strategy or processes being not properly represented in the model, uncertainties in the wind forcing are particularly relevant for AMOC estimates at 11° S.

## 1 Introduction

The Atlantic Meridional Overturning Circulation (AMOC) plays a major role in the global oceanic heat budget. Of the maximum heat transport of 1.3 PW found in the subtropical North Atlantic (e.g. Lavin et al., 1998) about 88% are carried by
the AMOC (Johns et al., 2011). Because of the AMOC, there is substantial northward heat transport across the Atlantic equator





(e.g. Talley, 2003), which is unique among global oceans. Simplifying the circulation in the Atlantic to a two-dimensional latitude-depth plane, the AMOC connects warm waters flowing northward in the upper ocean and cold waters flowing southward at depth across all latitudes by downward and upward motion, for example, in the subpolar North Atlantic or near the Southern Ocean (e.g. Buckley & Marshall, 2016). With the AMOC representing the strongest mode of northward heat

transport accomplished by the ocean, providing the observational evidence of the mechanisms that control its structure and variability is therefore of high priority for understanding the present-day climate, validating climate simulations and improving predictions.

Historically, the strength and structure of the AMOC was estimated based on shipboard hydrographic sections, which focused on establishing the mean AMOC strength and related heat transport (e.g. Richardson, 2008). The first trans-basin mooring

array - the RAPID/MOCHA transport array at 26° N - continuously measuring the temporal variability of the AMOC was set up in the early 2000s (Hirschi et al., 2003). From the observations at 26° N, we learned, that large AMOC variations can occur on a range of timescales - from weeks to decades (e.g. Srokosz & Bryden, 2015). Kanzow et al. (2007) showed, that not only the Ekman, but even more the geostrophic contribution to the AMOC, exhibit pronounced high-frequency variability with periods up to few weeks. With the identification of a strong seasonal cycle in the AMOC strength at 26° N, Kanzow et al.

(2010) demonstrated that AMOC estimates inferred from single hydrographic sections are subject to aliasing. They also found the upper-ocean geostrophic AMOC contribution to dominate on seasonal time scales, while Chidichimo et al. (2010) discovered those to be primarily driven by processes at the eastern boundary.

Today, there are several ongoing international efforts monitoring the AMOC at selected latitudes, such as - the OSNAP array in the subpolar North Atlantic (since 2014; Lozier et al., 2019), the RAPID array in the subtropical North Atlantic at 26° N

(since 2004; Cunningham et al., 2007; McCarthy et al., 2015), the MOVE array in the tropical North Atlantic at 16° N (since 2001; Kanzow et al., 2008; Send et al., 2011; Frajka-Williams et al., 2018), the SAMBA array in the subtropical South Atlantic at 34° S (since 2009; Meinen et al., 2018) – as well as other programs measuring important components of the overturning, such as - the Western Boundary Current (WBC) arrays at 53° N (since 1997; Zantopp et al. 2017) and at 39° N (Line W; 2004-2014; Toole et al., 2017), the array across the North Atlantic Current at 47° N (NOAC array; Roessler et al., 2015), the deep

overflow observations through Denmark Strait (Jochumsen et al., 2017) or Faroe Bank Channel (Hansen et al., 2016). In this study, we will present the first estimate of AMOC variations in the tropical South Atlantic - from the TRACOS (Tropical Atlantic Circulation and Overturning at 11° S) array.

The western tropical South Atlantic constitutes a key region for the exchange of water masses, heat and salt between the Southern and Northern Hemispheres (Biastoch et al., 2008a; Schmidtko & Johnson, 2012; Kolodziejczyk et al.,2014; Hummels

et al.,2015; Lübbecke et al., 2015; Herrford et al., 2017). Several observational and modelling studies (e.g. Rühs et al., 2016; Zhang et al., 2011) suggest that 11° S is a good place to monitor water mass signal propagation, changes in the WBC transports and, with that, changes in the AMOC transport. At 11° S the WBC regime is comprised of the northward North Brazil Undercurrent (NBUC) with a subsurface velocity maximum at about 200 m and the southward Deep Western Boundary Current (DWBC) below 1200 m (e.g. Schott et al., 2005). The NBUC is known to originate from the southern branch of the



South Equatorial Current (da Silveira et al., 1994), which transports subtropical waters towards Brazil and bifurcates between 14-28° S (Stramma and England 1999; Boebel et al. 1999; Wienders et al., 2000). Wind-driven variability in the South Atlantic subtropical gyre, and therefore, variations in the bifurcation latitude are thought to influence the strength of the NBUC farther downstream on different time scales (e.g. Rodrigues et al., 2007; Silva et al., 2009). From 2000 to 2004, a first mooring array was deployed at 11° S to observe the variability of the WBC and its components –the NBUC and the DWBC below. Schott et

al. (2005) found the NBUC to carry 25 Sv northward on average. The NBUC shows a strong seasonal cycle, which seems to be out of phase with the seasonal variations in the DWBC. Intraseasonal signals could also be observed: Dengler et al. (2004) described a spectral peak in the velocity time series at a period of 60-70 days, which could be observed in most of the moored records, but was strongest within the DWBC. They concluded that the DWBC transport at 11° S is mainly accomplished by migrating eddies. Further, Veleda et al. (2012) could relate variability at periods of 2-3 weeks to coastal trapped waves (CTWs)

propagating from 22-36° S equatorward along the Brazilian coast. In July 2013, a similar mooring array was again deployed at 11° S (Hummels et al., 2015), and is still in place. Comparing the two observational periods, Hummels et al. (2015) did not find significant changes in the averaged NBUC and DWBC transports. Furthermore, they could show that the interannual NBUC variability observed between 2000-2004 is consistent with the output of a forced ocean general circulation model (OGCM) named INALT01. Decadal variability in INALT01, however, was also found to be similar to transport estimates

based on historical hydrographic observations from Zhang et al. (2011). To date, Zhang et al. (2011) provide the only NBUC time series derived from hydrographic observations spanning several decades. They estimate multi-decadal variability of the NBUC to be of similar order as its seasonal cycle and, because of the connection to the Atlantic Multidecadal Variability, suggest the NBUC to serve as an index for AMOC variations on these time scales. In a model study, Rühs et al. (2016) found decadal to multi-decadal buoyancy-forced changes in the AMOC transport to manifest themselves in NBUC transport (at 6°

S), but also to be masked by interannual wind-driven variability.

With the resumption of the mooring array at 11° S in 2013, the observational program was also extended by installing a mooring array for direct velocity measurements across the continental slope off Angola. Studies based on these observations showed that the circulation there is weak and dominated by seasonal variability associated with remotely forced waves (Kopte et al., 2017; 2018). As shown in several model studies, most of the intraseasonal (T>120 days) to interannual variability in that

region is induced by a wave response to equatorial wind forcing that generates equatorial Kelvin waves propagating eastward and, while reaching the eastern boundary, transferring a part of their energy as CTWs further to the south towards 11° S (Illig et al., 2004; Illig et al., 2018; Bachèlery et al., 2016).

Besides the moored observations at 11° S, PIES (Pressure Inverted Echo Sounders) or single bottom pressure recorders (BPRs) were deployed on both sides of the Atlantic. Within some of the other programs targeting AMOC fluctuations, – such as

RAPID (Kanzow et al. 2010, McCarthy et al., 2015, Meinen et al. 2013), MOVE (Kanzow et al., 2006, 2008) and SAMBA (Meinen et al., 2017, 2018) – bottom pressure (BP) measurements are part of the "moored end point method", combining a time-varying reference velocity estimated from zonal BP differences with the internal geostrophic velocity derived from differences in dynamic height. Circulation changes in z-coordinates can even more straightforward be estimated by measuring



the pressure differences between the eastern and western boundary at each depth. In a model study, Bingham and Hughes

(2008) showed that this works well down to around 3000 m, even with only western boundary measurements. In our study, we use the BP differences across the basin at 300 m and 500 m depth to estimate the geostrophic contribution to AMOC variations in the tropical South Atlantic over the period 2013-2018 and investigate its seasonal variability in more detail.

## 2 Observational Data

### 1.1 Bottom pressure time series

Over the period 2013-2018 five BPRs were deployed at 11° S (Table 1). In July 2013, together with the WBC mooring array, two bottom-mounted PIES were installed across the Brazilian continental slope at 300 m and 500 m depth. PIES measure the acoustic travel time to the surface, as well as bottom pressure. In this study, we only use the BP time series. One year later another set of PIES was deployed at the same locations. While of the first set only the 500 m sensor could be recovered, the second set was maintained in September 2016 and spring 2018. Note, that the two PIES, KPO 1109 and KPO 1135 (Table 1)

were located only ~1 km away from each other over the period 05/2014 – 10/2015. At the eastern boundary off Angola, two SBE 26plus sensors (single or attached to an ADCP shield) measured pressure at 300 m and 500 m depth from 07/2013 to 11/2015. The instruments were re-deployed, but could not be recovered again. We assume, that they were lost due to extensive fishing in the region.

For our analyses, the available BP records are de-spiked, interpolated from an original sampling rate of 10 minutes to hourly

values and de-tided using harmonic fits with tidal periods shorter than 35 days. All tidal harmonics are calculated performing a classical harmonic analysis using the "Unified Tidal analysis and prediction" MATLAB® software (UTide; Codiga, 2011). The tidal models for T < 35 days capture between 97.0-99.6% of the total variance in the original BP time series. After removing these higher-frequency tides, the remaining variance is mainly related to seasonal variations and low-frequency instrument drifts. Instrument drifts vary substantially between the five instruments: While KPO 1106 shows almost no drift,

all other sensors exhibit a combination of exponential and linear behavior, but with different signs and at different rates (Fig. 2(a)). Unfortunately, we are not able to directly relate individual drift behavior to pressure effects or material creep. Earlier studies (e.g. Watts & Kontoyiannis, 1990; Johns et al., 2005; Kanzow et al., 2006; Cunningham et al, 2009) found subtracting a least-squares exponential-linear fit of the form $P_{Drift}(t) = a[1 - e^{-bt}] + ct + d$ from the pressure time series to be the procedure that works best for the PIES. As the SBE26plus recorders were also equipped with Quartz pressure sensors, we

decided to "de-drift" all five sensors similarly by subtracting exponential-linear fits as described above. Kanzow et al. (2006) also discussed the problem of this empirical de-drifting not being able to distinguish between the instrumental drift and ocean signals of the order of or longer than the time series. This means that, for example, seasonal signals can leak into the fit and its removal from the time series can reduce seasonal signals in return. We attempted to solve this problem by iteratively fitting an exponential-linear drift as well as annual and semi-annual harmonics. We started with a first guess of the exponential-linear

drift, removed it from the original time series and fitted annual and semi-annual harmonics to the de-drifted time series. The





first guess was improved in each iterative step by calculating new exponential-linear fits after subtracting the iteratively improved annual and semi-annual harmonics from the original data. After three repetitions the resulting fits tended to converge. Both fits from the third repetition are shown in Fig. 2(a). For further analyses, we removed the derived instrument drift from the original BP time series and averaged to daily values (Fig. 2(b)).

**1.2 Sea level anomalies**

To estimate pressure variability at the surface, we use sea level anomalies (SLA) from the delayed-time ''all-sat-merged'' data set of global sea surface height, produced by Ssalto/Duacs and provided by the Copernicus Marine Environment Monitoring Service (CMEMS). The multi-satellite altimeter sea surface heights are mapped on a 0.25° x 0.25° grid (e.g. Pujol et al., 2016) and are available for the period 1993-2018 at daily resolution. To obtain pressure variation near the boundaries, SLA grid

points are chosen closest to the Brazilian and Angolan coasts at 11° S, respectively. We tested the sensitivity of our results to SLA changes with distance to the coast (Fig. 2(c,d)): At the western boundary, off Brazil, the phase of the annual harmonic slightly changes with distance to the coast – about 30 days over 0.5° longitude. At the eastern boundary, off Angola, we found the phases of both annual and semi-annual harmonics to be constant over the distance between the location of the 300 m BPR and the coast.

**1.3 Wind stress**

In order to estimate the Ekman contribution to AMOC variability at 11° S we use gridded daily wind stress fields from Metop/ASCAT scatterometer retrievals. Those are available for the period 2007-2018 and with a spatial resolution of 0.25° x 0.25° (Bentamy & Fillon, 2012). The near-surface Ekman transport is estimated as the zonal integral of the zonal wind stress component between 10.5-11° S (see Eq. (7) in section 4.1).

**1.4 NBUC transport time series**

We show a transport time series of the NBUC, which is derived from four current meter moorings spanning the width of the NBUC at 11° S. Record gaps were filled with empirical orthogonal functions (EOFs) derived from the mooring data. Moored time series were finally mapped into sections every 2.5 days using a Gaussian-weighted interpolation with horizontal mapping scales of 20 km with a cutoff radius of 150 km and vertical mapping scales of 60 m with a cutoff radius of 1500 m (see

Hummels et al. (2015) for further details). The NBUC transport is calculated integrating the total flow (including northward and southward flow) within a predefined box bounded vertically by the surface and the neutral density surface $\gamma n$ =27.7 kg m-3, and horizontally by the Brazilian coast and the longitude 34.5° W. These transports represent updates from previous studies (Schott et al., 2005; Hummels et al., 2015).



## 3 Model Data

To validate the observational strategy, we use the 5-daily output from a hindcast experiment with the global ocean/sea-ice
Ocean General Circulation Model configuration `INALT01`. It is based on the NEMO (Nucleus for European Modelling of
the Ocean v3.1.1; Madec, 2008) code and developed within the DRAKKAR framework (The DRAKKAR Group, 2014).
INALT01 is a global 1/2° configuration with a 1/10° refinement between 70°W-70°E and 50°S-8°N, improving the
representation of the western boundary current regime in the South Atlantic and extended Agulhas region (Durgadoo et al.,

2013). It uses a tripolar horizontal grid, 46 vertical levels with increasing grid spacing and is forced by interannually varying
air-sea fluxes (1948-2007) from the CORE2b data set (Coordinated Ocean-ice Reference Experiments; Large & Yeager, 2009).
INALT01 uses the filtered free surface formulation for the surface pressure gradient - SSH is then a prognostic variable. This
particular model configuration has been previously used in the region. South of Africa it was used for validating a method of
determining Agulhas leakage from satellite altimetry (Le Bars et al., 2014). In the WBC region Hummels et al. (2015) found

interannual NBUC variability derived from moored observations and decadal NBUC variability from geostrophic estimates
(Zhang et al., 2011) to be consistent with the INALT01 output. Our analysis employs 2-dimensional (longitude-depth) sections
of temperature, salinity and velocity, as well as surface elevation fields along 11° S for the simulated period 1978-2007. Surface
wind stress fields are additionally shown for the years 2008-2009.

## 4 Methods

### 4.1 Computation of AMOC transport variations from BP observations


The structure of the AMOC is often described using the overturning transport stream function $\psi(y, z, t)$, which is derived from
integrating the meridional velocity component, v, zonally (from the eastern ($x_{EB}$) to the western boundary ($x_{WB}$)) and
accumulating it vertically:

$$\psi(y, z, t) = \int_{z}^{0} \int_{x_{WB}}^{x_{EB}} v(x, y, z', t) dx dz' \quad (1)$$

with x being longitude, y latitude, z the vertical coordinate pointing upward and t time. This reduces a complex three-
dimensional circulation system to a two-dimensional one. The AMOC strength or transport is commonly defined as the
maximum of $\psi$ over depth and typically expressed in Sverdrups [1 Sv = $10^6$ m$^3$ s$^{-1}$]. At any chosen latitude, $\Psi_{MAX}$ can be
decomposed into Ekman and geostrophic components:

$$\psi_{MAX}(y, t) = T_{AMOC}(y, t) = T_G(y, t) + T_{EK}(y, t) \quad (2)$$

Variations in the basin-wide upper-ocean meridional geostrophic transport $T_G$ at a certain latitude can be derived from the
differences between the bottom pressure at the eastern ($P_{EB}$) and western ($P_{WB}$) basin boundaries. At 11° S, we use bottom
pressure measurements on both sides of the basin at 300 m and 500 m depth. Figure 3 displays the observational strategy.

Unfortunately, technology limits our method as it is not possible to determine precisely the depth levels at which the instruments were placed with respect to equi-geopotential surfaces and, thus, only velocity anomalies can be determined (e.g.
Donohue et al., 2010). However, the differences between eastern and western boundary pressure anomalies from BPRs have successfully been used to estimate temporal fluctuations of the geostrophic contribution to AMOC variability (e.g. Kanzow et al., 2007).

At those depths, which are equipped with BPRs, anomalies of the geostrophic transport per unit depth $V'_G(z,t)$ are calculated as follows:

$$V'_G(z,t) = \frac{1}{\rho_0 \cdot f} \cdot \left(P'_{EB}(z,t) - P'_{WB}(z,t)\right) \quad (3)$$

$P'_{EB}$ and $P'_{WB}$ are the pressure anomalies at the eastern and western boundary with respect to the time mean, respectively, $f$ the Coriolis parameter and $\rho_0$ a mean sea water density. At the surface, $V'_G(z = 0, t)$ can be calculated from sea level anomalies, $\eta'$, at the eastern and western boundary:

$$V'_G(z = 0, t) = \frac{g}{f} \cdot \left(\eta'_{EB}(t) - \eta'_{WB}(t)\right) \quad (4)$$

with g being the acceleration of gravity. Additionally, a level of no motion is prescribed to be at 1130 m, such that $V'_G(z = -1130\ m, t) = 0$ at all times. This 'level of no motion' is estimated based on the velocity field from the INALT01 model configuration and defined as the depth of the local zero-crossing of v, averaged across the basin and over time. The maximum of the corresponding stream function is averaged over time located at z = -1072 m. Earlier studies, presenting transport estimates for 11° S, used a level-of-no-motion at the depths of $\sigma_1 = 32.15$ kg m$^{-3}$ (at about 1150 m; e.g. Stramma et al., 1995;
Schott et al., 2005).

We use two different methods to approximate the vertical structure of $V'_G$:

1.  Piecewise linear interpolation of $V'_G$ between the 4 data points at 0 m, 300 m, 500 m and 1130 m depth – denoted as $V'_{G\ Points}$ or $T'_{G\ Points}$ throughout the study.
2.  Regression of the 1st and 2nd EOFs, i.e. the two dominant vertical structure functions of the geostrophic transport per
unit depth derived from density and sea level anomalies in INALT01, $V'_{G\ SIM\ P(z)}$ (see chapter 4b), onto the 3 data points at 0 m, 300 m and 500 m depth thereby relaxing the no-flow condition at 1130 m depth. The first dominant vertical structure function explains 90.3% of the variance contained in $V'_{G\ SIM\ P(z)}$ and the second 9.6%. The transport variations resulting from this method are denoted as $V'_{G\ EOFs}$ or $T'_{G\ EOFs}$ in the following.

Upper-ocean geostrophic transport variations, $T'_G$, are then calculated by vertically integrating the approximated $V'_G$ profile
from $z_3$=-1130 m up to the surface. Using the first method, $z_3$ is defined as the 'level of no motion' in the simulated mean velocity field, separating the northward flow of warm Central Waters and Antarctic Intermediate Waters from the southward flow of deep waters below. For the second method, $z_3$ does not represent a 'level of no motion'.





$$T'_G(t) = \int_{z_3}^{0} V'_G(z,t)\ dz \quad (5)$$

Finally, AMOC transport variations ($T'_{AMOC}$) can be derived by adding local Ekman transport anomalies $T'_{EK}$.

$$T'_{AMOC}(t) = T'_G(t) + T'_{EK}(t) \quad (6)$$

Those can efficiently be estimated from the zonal component of the wind stress, $\tau_x$, at 11° S according to

$$T_{EK}(t) = -\int_{x_{WB}}^{X_{EB}} \frac{\tau_x(x,t)}{\rho_0 \cdot f}\ dx \quad (7)$$

and subtracting the temporal average.

The annual and semi-annual harmonics for all pressure time series are determined using the UTide-software (Codiga, 2011) and are presented together with an uncertainty estimate for their amplitudes in section 5a. The uncertainties for the amplitudes of the combined annual and semi-annual harmonics are derived by, first, low-pass filtering the pressure time series with a cutoff of 170 days, and then calculating the 95[th] percentile around the derived annual and semi-annual harmonics for every day of the year.

Following the observational strategy (Fig. 3), we would need at least four different locations at two depth levels to be equipped with BPRs in order to derive basin-wide geostrophic transport variations in the upper 1130 m of the water column. While we had five recorders in place over the period 15/05/2014-02/10/2015, there are no BP measurements at 300 m depth off Brazil before 05/2014 and none at all off Angola since 11/2015. In this study, we find seasonal variability to dominate all of the pressure time series at 11° S (see section 5.1). The combined annual and semi-annual cycles explain 44-61 % of the variance in the daily BP time series at the eastern boundary and 18-24 % of the variance at the western boundary. Therefore, we decided to "replace" the missing sensors with the combined annual and semi-annual harmonics derived from the available BP time series. This means, for example, that the geostrophic transport after 11/2015 is derived from the differences between measured BP variations at the western boundary and repeated annual and semi-annual harmonics – as derived from earlier years – at the eastern boundary.

**4.2 Using the OGCM INALT01 as a 'testing area'**

In order to validate our strategy for the computation of AMOC variations from the BP observations, but also to understand the observed seasonal variability, we simultaneously analyze the output of the OGCM INALT01 (see section 3).

In INALT01 we can diagnose AMOC variations, $T'_{AMOC\ SIM}$, directly from the velocity field using Eq. (1) and Eq. (2), i.e. by integrating the meridional velocity component at 11° S horizontally across the basin and vertically from 1130 m to the surface. The zonally integrated Ekman transport $T_{EK\ CORE2b}$ at 11° S is derived with Eq. (7) from CORE2b wind stress, which is used

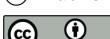



to force INALT01. According to Eq. (6) the simulated upper-ocean geostrophic transport anomaly $T'_{G\ SIM}$ is then the difference between $T'_{AMOC\ SIM}$ and $T'_{EK\ CORE2b}$.

Alternatively, we can also derive an estimate of the simulated upper-ocean geostrophic transport anomaly, $T'_{G\ SIM\ P(z)}$ by calculating BP at 11° S from the modelled hydrographic fields and sea level. The model pressure field is given by

$$p(x, z, t) = g \cdot \int_{z}^{0} \rho(x, z', t)\, dz' + g \cdot \rho_0 \cdot \eta(x, t) \quad (8)$$

with g being the acceleration of gravity, ρ the seawater density as function of z and η the sea level. Taking the BP along the continental slopes (at each depth level) of Brazil and Angola from Eq. (8), the simulated upper ocean geostrophic transport anomalies, $V'_{G\ SIM\ P(z)}$ and $T'_{G\ SIM\ P(z)}$, can be derived from the pressure differences across the basin using Eq. (3) and Eq. (5). Under the assumption, that ageostrophic non-Ekman velocities are negligible, $T'_{G\ SIM}$ and $T'_{G\ SIM\ P(z)}$ should agree and particularly should show the same seasonal cycles. We also test or observational strategy by applying the two methods used

to approximate the vertical structure of $V'_G$ from the observations (see section 4.1): 1. Piecewise linear interpolation between values of $V'_{G\ SIM\ P(z)}$ at 0 m, 300 m, 500 m depth and a level of no motion at 1130 m depth –denoted as $V'_{G\ SIM\ Points}$ or $T'_{G\ SIM\ Points}$ in the following. 2. Regression of the 1[st] and 2[nd] EOFs of $V'_{G\ SIM\ P(z)}$ onto the values $V'_{G\ SIM\ P(z)}$ at 0 m, 300 m, 500 m depth – deriving $V'_{G\ SIM\ EOFs}$ or $T'_{G\ SIM\ EOFs}$. These different transport estimates from INALT01 are used to validate the methods applied to the observations. As the resulting uncertainties are discussed mainly with regard to variations of the

seasonal cycle, the method test (particularly including the test of uncertainties due to the coarse vertical sampling at only few locations, i.e., at 300 m and 500 m depth) is part of section 5.3 presenting the seasonal variability of AMOC components. In section 5.4, we use INALT01 to identify relevant mechanisms of the seasonal AMOC variability at 11° S, including specifically a comparison of the seasonal variability of the NBUC transport derived from observations and INALT01. For the sake of simplicity, in INALT01, unlike for the calculations from observations, the NBUC transport is calculated above a fixed depth

of 1130 m and west of 34.55° W.

## 5 Results

### 5.1 Ocean pressure variability at 11°S

All of the ocean pressure time series we present in this study, i.e. at the surface from SLA (Fig. 2(a,b)), at 300 m and 500 m depth from the BPRs (Fig. 1(b)), at the western or eastern boundary, are dominated by seasonal variability. The corresponding

periodograms are shown as colored curves in Fig. 4 and all exhibit pronounced peaks at periods of the annual and semi-annual cycles.

Although we will mainly focus on seasonal variability in the following, it is worth to mention some other interesting peaks in the periodograms: There is energy on intraseasonal and interannual time scales. Off Brazil, variability at a period of 70 days



is very likely related to the DWBC eddies described by Dengler et al. (2004), which are thought to dominate the DWBC flow at 11° S and can influence the upper water column as well (e.g. Schott et al., 2005). The periodograms of SLA at the eastern boundary exhibit peaks at 90 days, 120 days and 2 years. Variability at periods of 90 days and 120 days were also observed by Kopte et al. (2018) in velocity time series from moored observations off Angola and are likely associated with the passage of CTWs. Interestingly, the OGCM INALT01 does reproduce the spectral peaks at 2 years, 120 days and 90 days in the SLA off Angola, but not the 70 days period observed in any of the BP time series.

We find the relative importance of seasonal variability to be most pronounced near the surface off Angola in both, the observations and the model (Fig. 4). The combined annual and semi-annual harmonics of the observed pressure time series explain most of the variance there – 61% at the surface, 58% at 300 m depth, 44% at 500 m depth – and their amplitudes decrease with depth. To make this statement we converted SLA variance into pressure variance using the hydrostatic equation. The seasonal cycle off Angola as reconstructed using the annual and semi-annual harmonics (Fig. 5(b,d,f)) shows a maximum in austral autumn, minimum in winter and second maximum in spring. But, the phases of the annual and semi-annual cycles

change with depth at different rates (Fig. 6). With a phase shift of about 5 months, the annual harmonics at the surface and 500m depth are almost out-of-phase. The semi-annual harmonic is rather in-phase peaking about 1.5 month earlier at depth. At the western boundary (Fig. 5(a,c,e)), the seasonal variability is less pronounced. The combined annual and semi-annual harmonics explain only 12% of the total variance at the surface and are barely different from zero, considering our uncertainty estimate for the amplitude. Seasonal variability of the surface pressure is decoupled from the pressure variability at depth. The

BP measurements at 300 m and 500 m depth, which are both located in depth range of the NBUC, have annual and semi-annual harmonics of similar amplitude and phase. The phase of the annual harmonic changes by 2 months between the surface and 300 m depth, the semi-annual harmonic by ~1 month and both peak later at depth. At depth, seasonal pressure variations also become more important - at 500 m depth, for example, the annual and semi-annual harmonics explain up to 29%.

Annual and semi-annual harmonics of the individual pressure time series simulated In the INALT01 model (grey shading in Fig. 5) agree quite well with the observations regarding the timing of the maxima and minima. On the other hand, there are large differences in the amplitudes: The model tends to overestimate the annual harmonic at the surface and underestimate seasonal variability in general at depth - especially at 300 m depth at the western boundary.

In this study, we estimate variations in the upper-ocean geostrophic transport as a contribution to the AMOC from pressure

differences across the basin. Based on the observations at 11° S, on seasonal time scales, we find eastern boundary pressure variations to prevail near the surface, whereas at 500 m depth the western and eastern boundary pressure variations are of similar importance. In the INALT01 model, the eastern boundary pressure variations dominate even more over western boundary ones.

**5.2 Wind stress variability**

Prevailing wind stress along 11° S is northwestward, which results in a mean southward Ekman transport. Using wind stress derived from ASCAT for the period 2013-2018 it is -11.7 ± 1.9 Sv and for the full available period 2007-2018 it is -11.8 ± 1.3





Sv. For the CORE2b wind stress used as forcing for the 30-year INALT01 model run, we derive an Ekman transport of -10.8 ± 0.5 Sv. While the zonal wind stress in the tropical South Atlantic varies on different time scales, it is clearly dominated by seasonal variability. Periodograms of the Ekman transport derived from ASCAT and CORE2b wind stress (Fig. 7(a,b) both

show the strongest peaks at the frequency of the annual cycle. The CORE2b winds do also show weak semi-annual variability, but only when considering the full time series from 1978-2009. Note, that the two products cover very different periods and that their periodograms both also hint towards longer-term variability, whenever considering the full available time series.

Comparing zonal wind stress anomalies at 11° S from the two analyzed wind products for the overlapping years 2008-2009 (Fig. 7(c,d)), we find both products to agree on the following aspects: Seasonal wind variability is clearly more pronounced in

the western part of the basin with zonal wind stress anomalies typically being westward (negative) in January to March – resulting in a weaker southward Ekman transport. In austral winter zonal wind stress anomalies are rather eastward (positive) and the southward Ekman transport is strongest - changing again towards the end of the year. For both wind products, the Ekman transport across 11° S is mainly governed by the seasonal cycle of the southeasterly trade winds (e.g. Philander & Pacanowski, 1986). However, there are also recognizable differences between both products: For 2008-2009, the mean and

the monthly standard deviation of the Ekman transport at 11° S are about 0.5 Sv larger for ASCAT than for CORE2b, respectively. Wind stress anomalies along 11° S reveal differences in its spatial structure, as well as in the course and amplitudes of its seasonal cycle (Fig. 7(c,d)).

### 5.3 Seasonal variability of the AMOC components at 11°S

As described in the methods, we are able to estimate AMOC transport variations in the tropical South Atlantic from BP

measurements over the period 2013-2018. Figure 8 displays the derived time series of $T_G'$, $T_{EK}'$ and, being the sum of both components, $T_{AMOC}'$ at 11° S. The different versions of $T_G'$ derived from 4 BPRs or from 2-3 BPRs complemented with the annual and semi-annual harmonics from the fully equipped period (Fig. 8(a), see section 4.1) show a general good agreement within the overlapping period. In the following, we will use only a combined timeseries derived from all available BPR time series.

From the BP observations we can only derive anomalies of $T_G$. In INALT01, on the other hand, we can calculate also mean values: The AMOC transport at 11° S calculated from the velocity field in INALT01 averaged over the whole model run (1978-2007) is $T_{AMOC\ SIM}$ = 14.1 ± 0.9 Sv. This is within the uncertainty range of the AMOC estimate of 16.2 ± 3.0 Sv derived from a hydrographic ship section along 11° S in 1994 (Lumpkin & Speer, 2007).

Both, the $T_G'$ and $T_{EK}'$ time series show variability on different time scales, but are clearly dominated by seasonal variability

and, correspondingly, $T_{AMOC}'$ is as well. The mean seasonal cycles of $T_G'$, $T_{EK}'$ and $T_{AMOC}'$ from observations for the period 2013-2018, as well as from INALT01, are shown in Fig. 9. Despite having different products and different periods, we find the timing of the seasonal cycles in the observations and in INALT01 to be very similar for all components – especially, when considering possible interannual variations of the seasonal cycle.



$T'_{EK}$ is characterized by a maximum southward transport in June-August and minimum southward transport in January-March,

with the individual extrema slightly varying between ASCAT and CORE2b. Note again, that both products are averaged over different periods. The peak-to-peak amplitude of seasonal Ekman transport variations is 7.1 Sv for ASCAT winds over the period 2007-2018 (Fig. 9(c)) and 4.9 Sv for CORE2b winds for 1978-2009 (Fig. 9(d)). Section 5.2 describes the differences in wind stress between both products in more detail. We can give an estimate of the interannual variability by showing the range of mean seasonal cycles calculated for running 5-year subsets of the available wind stress data. While the timing of the

seasonal cycle of $T'_{EK}$ does not change much between different periods, its amplitude changes, showing that peak-to-peak amplitudes of 6-11 Sv for ASCAT and 2-8 Sv for CORE2b are possible.

The observed upper-ocean geostrophic transport shows a maximum in June, while minima occur in October and January with a weak secondary maximum in between in December. The two estimates, $T'_{G\ Points}$ and $T'_{G\ EOFs}$, derived with two different methods, agree well in the timing of minima and maxima (Fig. 9(a)). However, the amplitude of the seasonal cycle of $T'_{G\ EOFs}$,

which we will be considered as the more realistic solution in the following, is about 2 Sv smaller than the corresponding amplitude of $T'_{G\ Points}$. A possible explanation for the difference between the two estimates based on observations will be given below when discussing the vertical structure of the geostrophic transport per unit depth from simulations (Fig. 10).

Nevertheless, the seasonal cycles of both estimates, $T'_{G\ Points}$ and $T'_{G\ EOFs}$, are substantially more pronounced than that of $T'_{G\ SIM}$ derived from the velocity fields of the 30-year model run (Fig. 9(b)). The peak-to-peak amplitude of the seasonal cycle of

$T'_{G\ SIM}$ averaged over 30 years is 5.5 Sv, while amplitude ranges between 2-10 Sv for 5-year subsets. The peak-to-peak amplitude of $T'_{G\ EOFs}$ is 12.2 Sv averaged over the observed 4.5 years, or between 10.6-14.2 Sv considering the standard errors, which is just out of the range of simulated values. Regarding the timing of minima and maxima, the observed and simulated seasonal cycles of $T'_G$ agree well (cf. Fig. 9(a,b)). The larger peak-to-peak amplitudes of the seasonal cycle of the observed upper-ocean geostrophic as well as the ASCAT Ekman transports result in a larger seasonal cycle of $T'_{AMOC}$ compared to

$T'_{AMOC\ SIM}$.

Testing our observational strategy in INALT01, we compare the upper-ocean geostrophic transport anomaly derived from the simulated meridional velocity component ($T'_{G\ SIM}$) to that one being derived from simulated BP time series. $V'_{G\ SIM\ P(z)}$ and $T'_{G\ SIM\ P(z)}$ (Fig. 10(a,b)) are calculated using the full vertical resolution of the model and, as expected, $T'_{G\ SIM\ P(z)}$ agrees well with $T'_{G\ SIM}$. To resemble the observations, we also use piecewise linear interpolation between the depths of the pressure

observations at 0 m, 300 m, 500 m depth and the 'level of no motion' at 1130m ($V'_{G\ SIM\ Points}$; $T'_{G\ SIM\ Points}$; Fig. 10(c,d)). In INALT01, we find this method to miss certain parts of the vertical structure of $V'_{G\ SIM\ P(z)}$, and with that, to substantially overestimate the peak-to-peak amplitude of the seasonal cycle of $T'_{G\ SIM\ P(z)}$ by 6 Sv (Fig. 10(d)). While in the model a strong seasonal cycle is confined to the near-surface ocean, using linear interpolation between the surface and 300m artificially increases the seasonal signal in the layer from 50 to 250 m depth. Another method is based on a regression of the 1st and 2nd

dominant vertical structure functions of $V'_{G\ SIM\ P(z)}$ onto the values at the 3 depth levels of pressure observations at 0 m, 300 m, 500 m depth ($T'_{G\ SIM\ EOFs}$; $V'_{G\ SIM\ EOFs}$, Fig. 10(e,f)) thereby relaxing the no-flow condition at 1130 m depth. As the first two



EOFs of $V'_{G\,SIM\,P(z)}$ explain 99% of the variance contained in $V'_{G\,SIM\,P(z)}$, $T'_{G\,SIM\,EOFs}$ agrees well with $T'_{G\,SIM\,P(z)}$ in INALT01 (Fig. 10(f)). However, from comparing the observed BP time series with the BP simulated in INALT01, we learned that the model tends to underestimate the seasonal pressure variability at depth (see section 5.1) leaving some uncertainty regarding

the vertical structure of $V'_G$ in reality.

Figure 11 compares the mean seasonal cycles of $V'_G$ from observations for the two different methods. Using the vertical structure from the EOFs of $V'_{G\,SIM\,P(Z)}$ from INALT01 does especially reduce the amplitude of the subsurface variability (50-200 m). In this depth range the transition from negative to positive transport anomalies also shifts from April to March. At larger depths, differences between both methods result from the fact that $V'_{G\,Points}$ is constraint by a level of no motion at 1130

m, while $V'_{G\,EOFs}$ is not. However, independent of the applied method, the peak-to-peak amplitude of the seasonal cycle of $T'_G$ from observations (Fig. 9(a)) remains to be substantially larger than that from INALT01.

This means, that for the period 2013-2018, the geostrophic contribution to the seasonal cycle of the AMOC at 11° S, as we observe it, exceeds the Ekman contribution almost by a factor of 2 (cf. Fig. 9(a,c)). In INALT01, on the other hand, averaged over the 30-year model run, the geostrophic and Ekman contributions are of similar magnitude (Fig. 9(b,d)). The seasonal

cycles of both contributions vary substantially between years – e.g. 2-10 Sv for $T'_{G\,SIM}$ from INALT01, 2-8 Sv for $T'_{EK}$ from CORE2b or 6-11 Sv for $T'_{EK}$ from ASCAT - which means that there is a modulation of the ratios of both contributions on interannual time scales. However, even when considering the standard errors calculated for the seasonal cycle of $T'_{G\,Points}$ or $T'_{G\,EOFs}$ over 2013-2018 (Fig. 9(a)) and the range of mean seasonal cycles of $T'_{G\,SIM}$ calculated for 5-year subsets of the model run period 1978-2007 (Fig. 9(b)), the observed values are significantly larger than simulated ones.

**5.4 Dynamics of the seasonal cycle at 11°S**

In order to better understand the mechanisms that set the seasonal cycle of $T'_{AMOC}$ at 11° S, we investigate the longitudinal structure of the geostrophic velocity field and the geostrophic transport along that section in INALT01. We are able to distinguish three different regimes – the NBUC, the Western Basin Interior and the Eastern Basin - all showing seasonal variability of similar magnitude (Fig. 12).

The mean seasonal cycle of the NBUC, as calculated for the 30-year INALT01 model run, has its maximum in April, minimum in November and a peak-to-peak amplitude of 10 Sv (Fig. 12(b)). Peak-to-peak amplitudes of up to 15 Sv are found in 5-year subsets of the model time series. Having a mooring array installed off the coast off Brazil measuring the Western Boundary Current system there (Schott et al., 2005; Hummels et al., 2015), allows us to directly compare the seasonal variability of the NBUC in INALT01 with observations. The seasonal cycle of the NBUC in INALT01 agrees quite well with the seasonal cycle

observed in recent years – regarding the peak-to-peak amplitude (7.6 Sv in 2000-2004 and 7 Sv in 2013-2018) and the timing of maximum and minimum transports (Fig. 13(b)). During the earlier deployment period 2000-2004 there is a stronger semi-annual cycle creating a secondary minimum in March, which is neither found in the observations during 2013-2018 nor in INALT01.



In INALT01, the contribution of the NBUC to the AMOC on seasonal time scales is largely compensated by the flow in the
western basin interior. The seasonal cycle of the geostrophic transport per unit depth in the western basin interior is of similar
strength and vertical structure, but opposing sign to the one of the NBUC (cf. Fig. 12(a,c)). In the western basin interior, the
vertically integrated upper-ocean geostrophic velocity is mainly associated with an annual harmonic and likely related to a
strong seasonal cycle in the local wind stress curl (Fig. 14(a)). The annual harmonic of the wind stress curl exhibits relatively
large amplitudes over the region (15° W to 34.55° W) and a westward phase propagation (not shown). In the same region,
below the Ekman layer, the phases of the annual harmonic of the meridional velocity show a low baroclinic mode structure
(Fig. 14(b)).

As the contributions of the NBUC and western basin interior seasonal cycles to the AMOC tend to cancel out each other, in
INALT01, seasonal variability of the upper-ocean geostrophic transport at 11° S is mainly set in the eastern basin (Fig. 12(f)).
Away from the Ekman layer, the annual and semi-annual harmonics of the meridional velocity in the eastern basin are
dominated by westward and upward phase propagation (Fig. 14(b,c)). This suggests the existence of westward and downward
propagating Rossby beams likely related to remote equatorial forcing (e.g. Chu et al., 2007).

In addition to a westward, upward propagation, the phases of the annual and semi-annual harmonics of the meridional velocity
in the eastern basin also show a vertical structure along the continental slope and sustained into the eastern basin interior. In
INALT01, Rossby waves dominate the meridional velocity east of 5° W, but largely do not cross the Mid-Atlantic Ridge into
the western basin indicated by the minimum in the annual and semi-annual harmonics in mid-basin (Fig. 14).

From this analysis, we conclude that the seasonal variability of the geostrophic contribution to the AMOC at 11° S mainly
results from the oceanic adjustment to local and remote wind forcing. In INALT01, a compensation between the NBUC and
western basin interior results in a minor contribution of the upper-ocean geostrophic transport from the western basin to the
AMOC transport compared to a major contribution from the eastern basin on seasonal time scales. As described in section 5.1,
however, the model tends to underestimate the seasonal pressure variability at 300 m and 500 m depth – especially at the
western boundary. This leaves some uncertainty in the relative importance of western and eastern basin contributions to total
AMOC variability in reality.

## 6 Summary and discussion

In this study, we use bottom pressure observations on both sides of the basin at 300 m and 500 m depth, combined with satellite
measurements of sea level anomalies, different wind stress products and model results, to estimate the upper-ocean geostrophic
transport contribution to AMOC variability at 11° S over the period 2013-2018.

The use of bottom pressure measurements to compute basin-wide integrated northward transports is not straightforward:
Firstly, the sensors experience instrumental drifts, which limits the BPRs capabilities to recover variability on longer time
scales. Secondly, the deployment depth is not precisely known, which allows only the calculation of transport anomalies, but



not of mean values. We find the available BP time series at 11° S to be just long enough to investigate the seasonal variability
        in that region.

        At 11° S, seasonal variability is strong in all of the time series presented in this study. After removing tides with periods shorter
        than 35 days, the combined annual and semi-annual harmonics explain most of the variability at the eastern boundary – from
        60% at the surface to 44% at 500 m depth. We find hints towards a baroclinic structure in the annual and semi-annual harmonics
of the pressure time series (Fig. 6), which could be related to CTWs of specific baroclinic mode. The observed annual harmonic
        of the alongshore velocity at 11° S presented in Kopte et al. (2018) also exhibits a flow reversal between the surface and ~50
        m, while the semi-annual harmonic is instead characterized by a more gradual change with depth. They confirm the association
        of the annual and semi-annual cycles with basin-mode resonance in the equatorial Atlantic of the fourth and second baroclinic
        modes, respectively (Brandt et al., 2016). CTWs of this baroclinic structure can travel along the African coast towards 11° S
thereby impacting the local velocity and pressure fields.

        The region off Angola is known to be dominated by CTWs not only on seasonal, but on intraseasonal to interannual time scales
        (e.g. Bachèlery et al., 2016; Illig et al., 2018a,b; Kopte et al., 2018). The periodograms of SLA at the eastern boundary exhibit
        peaks at 90 days and 120 days. The peaks at periods of 90 days and 120 days have also been found by Kopte et al. (2018) in
        velocity time series from moored observations at 11° S off Angola. Based on numerical experiments, Bachèlery et al. (2016)
showed that SLA variability along the African coast is on intraseasonal time scales (T<105 days) primarily driven by local
        atmospheric forcing, while energetic signals at periods >120 days are mostly explained by equatorial forcing. Further, Polo et
        al. (2008) suggested that the intraseasonal variability is related to year-to-year variations of the seasonal cycle. Note that there
        is enhanced spectral energy at periods of 1.5 - 2.5 years in the eastern boundary SLA, which is also present when considering
        the full time series from 1993-2018 (Fig. 4(b)).

At the western boundary, seasonal pressure variability is weaker, but its relative importance compared to other variability
        increases with depth - the annual and semi-annual harmonics explain about 10% of the variability at the surface and 30% at
        500 m depth. The seasonal variability of the zonally integrated geostrophic velocity anomaly in the upper 300 m is, therefore,
        mainly controlled by pressure variations at the eastern boundary, while at 500 m depth contributions from the western and
        eastern boundaries are similar.

Annual und semi-annual harmonics at the western boundary also exhibit a vertical structure as seasonal variability at the
        surface is decoupled from the pressure variability at 300 m and 500 m depth. Based on geostrophic velocity fields from
        hydrographic measurements, studies like Silveira et al. (1994) or Stramma et al. (1995) already stated that the WBC system at
        11° S includes a pronounced undercurrent, the NBUC, above which higher current variability was observed. Further, the BP
        periodograms show a peak at 70 days (Fig. 4(c,e)), which is likely related to the DWBC eddies described by Dengler et al.
(2004) also influencing the upper water column.



Over the period 2013-2018, the upper-ocean geostrophic transport variations derived from pressure differences across the basin, are dominated by seasonal variability – with a peak-to-peak amplitude of 12-14 Sv, depending on the method used to approximate its vertical structure. The peak-to-peak amplitude of the mean seasonal cycle of the Ekman transport is 7 Sv and of the resulting AMOC transport 14-16 Sv. From moored observations of the Western Boundary Current system off Brazil

(updated from Schott et al., 2005; Hummels et al., 2015), we obtain a peak-to-peak seasonal variability of the NBUC of 7 Sv. The observed seasonal cycles of the AMOC, its components as well as the NBUC are in general good agreement with the output of the OGCM INALT01.

Nevertheless, there are some notable differences:

- The INALT01 model tends to underestimate the seasonal pressure variability at depth (300 m and 500 m), especially
at the western boundary. This translates into the vertical structure of the simulated geostrophic transport variations, which is used for the calculation of the observational estimate and, therefore, adds to its uncertainty.

- In the observations, the geostrophic contribution to seasonal AMOC variability exceeds the Ekman contribution by almost a factor of 2, while in INALT01, averaged over the 30-year model run, the contributions are similar. About equal geostrophic and Ekman contributions were also observed at other latitudes (at 26° N in Kanzow et al., 2010; at
34.5° S in Dong et al., 2014) or identified in earlier studies based on models (e.g. Zhao & Johns, 2014). Even when considering the standard errors calculated for the seasonal cycle of $T'_G$ over 2013-2018 (Fig. 9(a)) and the range of mean seasonal cycles of $T'_{G\ SIM}$ calculated for 5-year subsets of the model run period 1978-2007 (Fig. 9(b)), the observed values are significantly larger than the simulated values.

- The ratios of the NBUC and AMOC seasonal amplitudes are different between the observations (<1) and the model
(>1).

In the model, seasonal AMOC variability at 11° S is governed by the variability in the eastern basin, where the annual and semi-annual harmonics of the meridional velocity is dominated by westward and upward phase propagation (Fig. 14(b,c)). This can be related to equatorial forcing resulting finally into long Rossby waves to be radiated from the Angolan continental slope and travelling westward into the basin interior. Similar Rossby-wave radiation from the eastern boundary have been

reported for the tropical North Atlantic (e.g. Chu et al., 2007) and proposed to be one of the main mechanisms for seasonal variations in the geostrophic transport there (e.g. Hirschi et al., 2006; Zhao & Johns, 2014).

In addition to a westward propagation, the phases of the annual and semi-annual harmonics of the meridional velocity in the eastern basin also show a specific vertical structure. We interpret this to be related to the baroclinic structures of seasonal CTWs (e.g. fourth mode for annual cycle; Brandt et al., 2016; Kopte et al., 2018) arriving at 11° S, being radiated as Rossby

waves with a similar baroclinic structure.

Concurrently, the seasonal variability of the NBUC is mainly compensated by a low baroclinic mode signal in the western basin interior. Being related to a strong seasonal cycle of the wind stress curl in the western basin, we interpret this signal as



locally forced Rossby waves. In an early model study, Döös (1999) described the influence of long Rossby waves on the seasonal cycle in the western tropical Atlantic. Also, Schott et al. (2005) suggested the seasonal cycles of the NBUC and
DWBC at 11° S to be a response to interior wind forcing, occurring as a first baroclinic mode Rossby wave. Locally wind-forced seasonal Rossby waves would travel westward and after arriving at the western boundary directly force WBC variability.

The compensation between the western basin interior and the NBUC on seasonal time scales found in INALT01 results in a minor contribution of the western basin compared to the contribution from the eastern basin to and limits the importance of
the NBUC for AMOC variability on seasonal time scales. But, it is also subject to numerous uncertainties: In this study we found that INALT01 tends to underestimate seasonal variability in the central water layers off Brazil. In two different model studies, Rodrigues et al. (2007) and Silva et al. (2009), related seasonal variability in the NBUC to seasonal variations in the bifurcation region of the South Equatorial Current. Thus, the phases of the annual and semi-annual harmonics of the NBUC may not be simply set by the response to the local wind curl forcing in the western basin at 11° S, they may also depend on
the wind curl forcing farther south and associated equatorward signal propagation along the western boundary.
We conclude that the seasonal variability of the geostrophic contribution to the AMOC at 11° S is mainly wind-forced, as it is modulated by oceanic adjustment to local and remote wind forcing. While some of the uncertainties of our analysis result from the technical aspects of the observational strategy or processes being not properly represented in the model, our results indicate, that uncertainties in the wind forcing are particularly relevant for AMOC estimates in the tropical South Atlantic. Differences
between wind products are an important source of uncertainty for estimates of the AMOC and its variability. Especially, when comparing estimates of AMOC strength and variability between different projects, latitudes or from observations and models, the choice of wind product is crucial.
This study adds to the overall understanding of local and shorter-term AMOC variations, which is important for estimating the significance of long-term AMOC changes and, thus, for the detectability of its meridional coherence. To predict the long-term
behavior of the AMOC and its impacts, continuous observations from purposefully designed arrays are needed in different key locations. We would like to argue that the observational program at 11° S, if continued into the future, has potential for monitoring long-term AMOC changes. The western tropical Atlantic being a crossroad for the different branches of the AMOC and a region with high signal-to-noise ratios, makes 11° S a good place to monitor AMOC variations. Having a sustainable AMOC observing system there, linking northern and southern AMOC variability, would contribute to the general
understanding of related mechanisms. There is potential in using the setup of the BPRs for investigating longer-term AMOC variability. While progress is made in solving the problems of bottom pressure sensors on longer time scales (e.g. Kajikawa & Kobata, 2014; Worthington et al., 2019), the advantage of our method is that the BPRs are less expensive and easier to deploy than full-height mooring arrays. Additionally, we can fall back on more than 20 years of shipboard hydrographic measurements in the tropical South Atlantic – at the western (e.g. Hummels et al., 2015; Herrford et al., 2017) and eastern boundary (e.g.



Tchipalanga et al., 2018). Ongoing work includes combining all of these hydrographic measurements to extend the time series of the WBC system and AMOC at 11° S back into the 1990s.

**Author contribution**

The methodology was first proposed T.K., then further developed and conceptualized by J.H. and P.B. P.B. and R.H. raised the project funding and, together with M.A., administered the project. The investigation was made by J.H., supervised and

validated by P.B. and T.K. J.H. processed the observational data, performed all analyses, drafted the manuscript and designed the figures. J.V.D. developed INALT01 and performed the simulations. R.H. calculated and provided the NBUC transport time series. All authors contributed to the discussion of the results or the review and editing of the manuscript.

**Disclaimer**

The authors declare that they have no conflict of interest.

**Acknowledgements**

This study was funded by the Deutsche Bundesministerium für Bildung und Forschung (BMBF) as part of the projects RACE (03F0651B, 03F0729C, 03F0824C), SACUS (03G0837A) and BANINO (03F0795A, 03F0795C), by the EU H2020 under grant agreement 817578 TRIATLAS project and by the Deutsche Forschungsgemeinschaft (DFG) through funding of Meteor cruises. We thank captains and crews of the R/V Meteor and R/V Sonne, as well as our technicians for the assistance during

the shipboard and moored station work. Data sets described in section 2.1 are available through https://doi.pangaea.de/ SLA data was distributed by the E.U. Copernicus Marine Service Information. INALT01 was developed at GEOMAR, with details of its configuration and access to data available at www.geomar.de/en/research/fb1/fb1-od/ocean-models/inalt01/. J.V.D acknowledges funding from the Helmholtz Association and the GEOMAR Helmholtz Centre for Ocean Research Kiel (grant IV014/GH018). We would like to thank G. Krahmann and M. Dengler for helpful discussions.

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





| Acronym | Mooring ID | Instrument | Position | Depth | Deployment period |
|---|---|---|---|---|---|
| P WB 500m a | KPO 1109 | PIES | 10.2367°S 35.8633°W | 500m | 05/2013 - 10/2015 |
| P EB 300m | KPO 1110 | Single SBE 26plus sensor | 10.6830°S 13.2250°E | 300m | 07/2013 - 11/2015 |
| P EB 500m | KPO 1106 | ADCP shield with SBE 26plus sensor | 10.7090°S 13.1855°E | 500m | 07/2013 - 10/2015 |
| P WB 300m | KPO 1134 | PIES | 10.2320°S 35.8780°W | 300m | 05/2014 - 09/2016 09/2016 - 03/2018* |
| P WB 500m b | KPO 1135 | PIES | 10.2430°S 35.8700°W | 500m | 05/2014 - 09/2016 09/2016 - 02/2018* |

**Table 1 Collection of available BP measurements at 11° S. Acronyms used throughout this article are given in the 1st column, official mooring IDs and instrument types are listed in the 2nd and 3rd columns. Columns 4-6 give the positions, depths and deployment periods for each BP measurement. The BP data can be found at https://doi.pangaea.de/10.1594/ PANGAEA.907589. *These sensors were re-deployed in 2018 and are currently in place.**





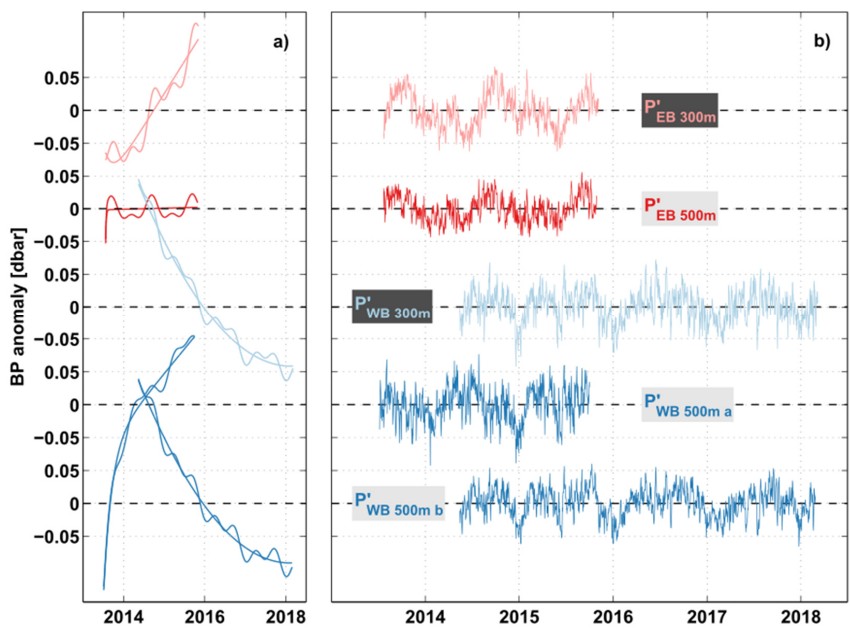

**Figure 1 Time series of BP anomalies measured at 11° S off Angola at 300 m (pink) and 500 m (red), as well as off Brazil at 300 m (light blue) and 500 m (blue) depth. a) Instrument drifts that are removed from the individual BP anomaly time series shown in b), as well as the sum of the drift and the combined annual and semi-annual harmonics. b) Daily time series of BP anomalies after handling, de-tiding and de-drifting (see text for details).**



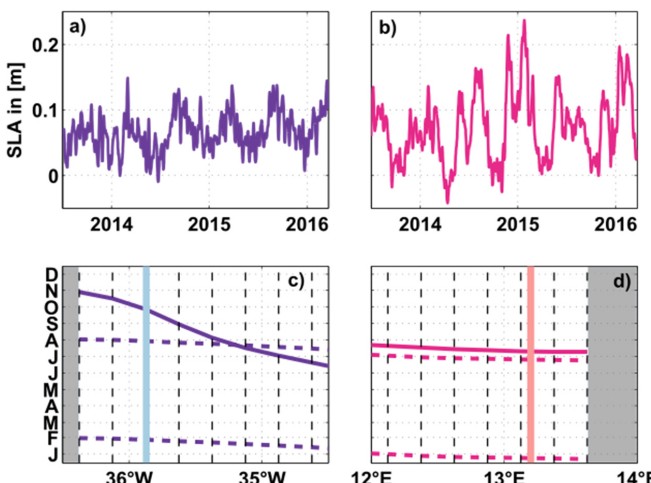

**Figure 2 (a,b) Time series of SLA over the period 2013-2018 – chosen close to the western boundary (purple) and eastern boundary (magenta). (c, d) Phases of the minima of the annual (solid curve) and semi-annual (dashed curves) harmonics. Black dashed lines represent the zonal grid spacing of the SLA data. Light blue (pink) lines mark the locations of the 300 m BPRs at the western (eastern) boundary. Gray areas mark land.**


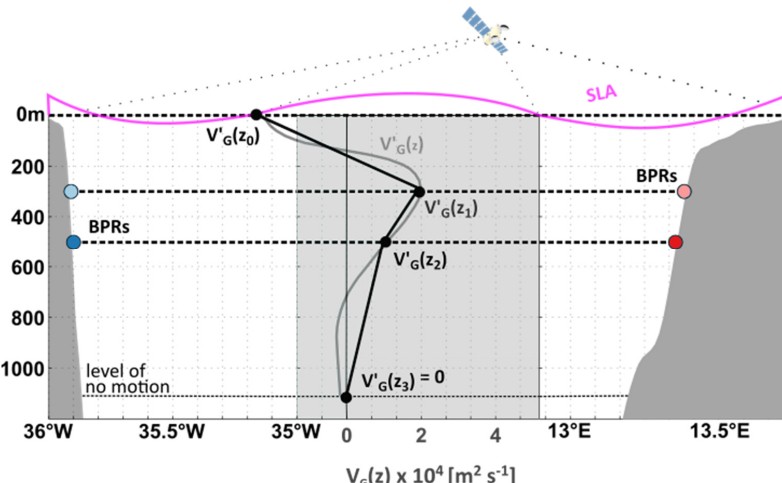

**Figure 3 Experimental setup and strategy to estimate** $T'_G$ **showing the location of the BPRs (red & blue circles) and the vertical sampling of** $V'_G$**.** $V'_G$ **is derived from measurements of sea level anomaly, z0=0 m, and with BPRs at two depth levels, z1=-300 m and z2=-500 m. A level of no motion is prescribed to be at z3=-1130 m. To approximate** $V'_G$ **two methods are used: i) piecewise linear**
**interpolation of** $V'_G$ **between the 4 data points (black profile), ii) regression of the 1st and 2nd dominant vertical structure functions of** $V'_{G\ SIM\ P(z)}$ **from INALT01 onto the data points at 0 m, 300 m and 500 m depth relaxing the no-flow condition at 1130 m depth (grey profile).** $V'_G$ **is then vertically integrated to derive** $T'_G$**.**


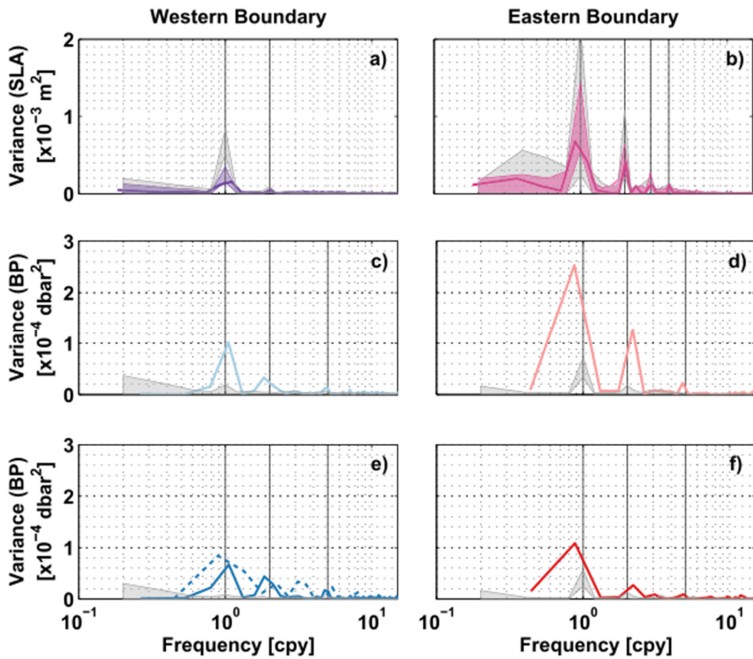

**Figure 4 Periodogram of (a-b) SLA, (c-d) BP at 300 m and (e-f) BP at 500 m depth from observations (colored lines) and from the**
**INALT01 model (grey shading). Frequency is given in 'cycles per year'. Bold solid colored curves show periodograms calculated from the available data for the period 2013-2018. In (a-b) thinner colored curves give the minimum and maximum ranges of periodograms calculated for 5-year windows running through the whole period of available SLA data (1993-2018). In e) the blue solid curve shows the spectrum for KPO 1135 and the blue dashed curve for KPO 1109 (see Table 1 for observational periods). Grey shading gives the minimum and maximum ranges of periodograms for pressure time series derived from the INALT01 model, again**
**calculated for 5-year windows running through the period of available data (1978-2007). Black vertical lines mark the frequencies of the annual and semi-annual cycles, as well as periods of 120 and 90 days in (b) or 70 days in (c-f).**


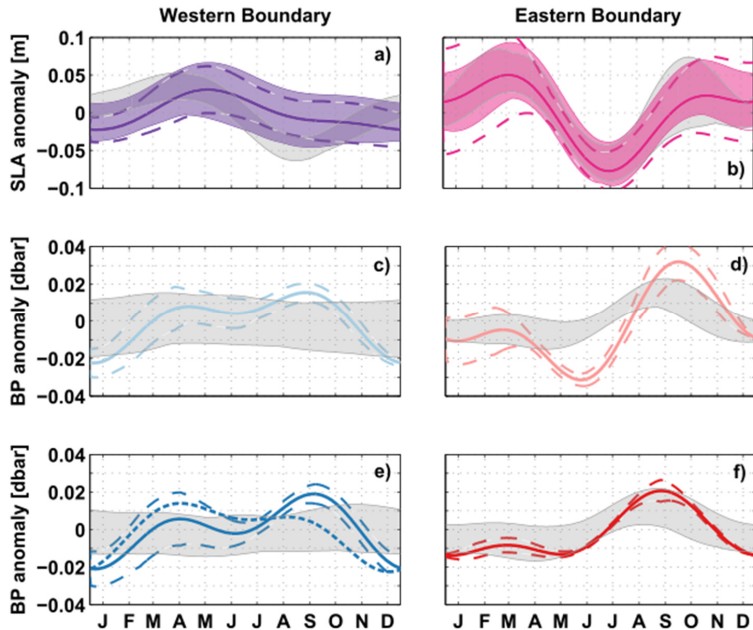

**Figure 5 Combined annual and semi-annual harmonics calculated for (a-b) SLA, (c-d) BP at 300 m and (e-f) BP at 500 m depth. Line styles and color coding are the same as in Fig. 4. Additionally, dashed lines and curves give uncertainties for the amplitudes of the harmonics calculated as described in section 4.1.**






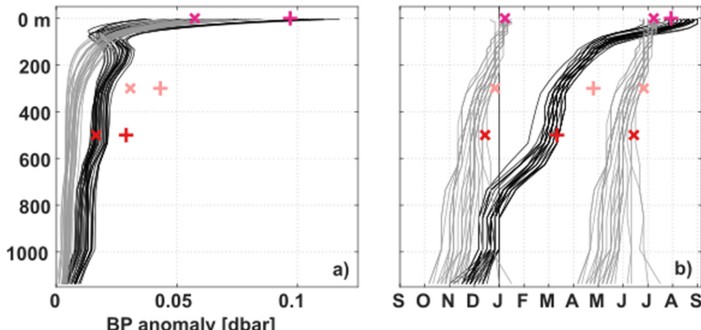

**Figure 6 (a) Amplitudes and (b) phases of the annual (pluses and black curves) and semi-annual (crosses and grey curves) harmonics**
**of the pressure anomalies at the eastern boundary along 11° S. Markers represent estimates from the observations (2013-2018) at**
**0m (magenta), at 300m (pink), 500m (red), the curves show estimates calculated from INALT01 for 5-year windows running through**
**the period of available data (1978-2007).**

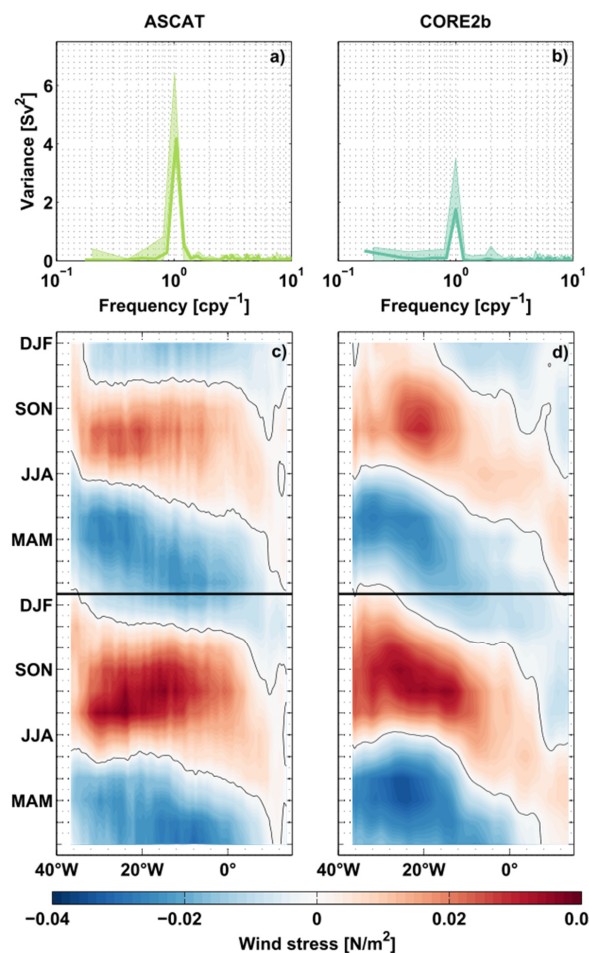

**Figure 7 (a-b) Periodogram of the Ekman transport at 11° S. Bold curves are calculated for ASCAT (left panels) over the period**
**2013-2018 and for CORE2b (right panels) over 2002-2007. Thin curves give the minimum and maximum ranges of periodograms calculated for 5-year windows running through the full time series - 1993-2018 for ASCAT and 1978-2009 for CORE2b. Frequency is given in 'cycles per year'. (c-d) Hovmoeller diagrams of the ASCAT and CORE2b zonal wind stress anomalies along 11° S for the overlapping years 2008-2009. Red (blue) colors imply eastward (westward) wind stress anomalies.**


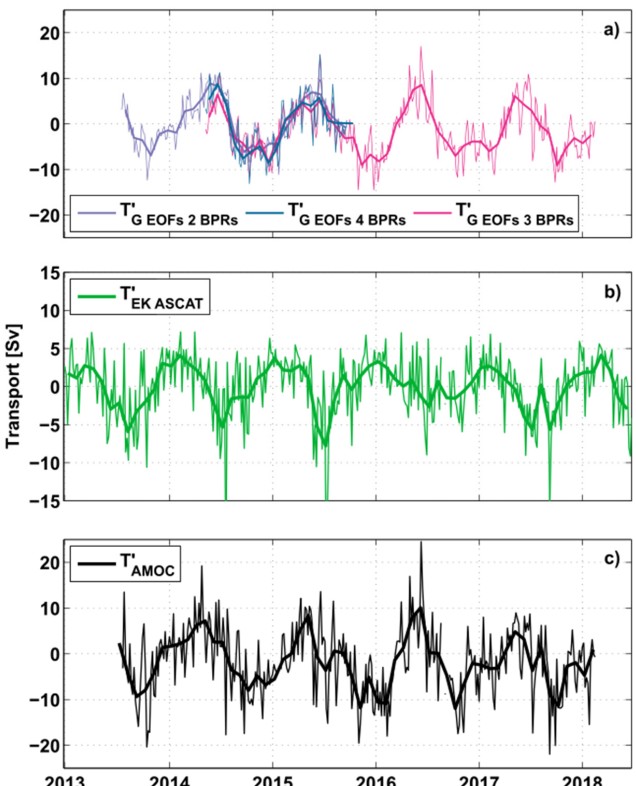

**Figure 8 Anomaly time series at 11° S (daily in a) and 5-daily in (b-c) as thin lines, and monthly averages as bold lines) of (a) the upper-ocean geostrophic transport ($T'_{G\ EOFs}$), (b) the Ekman transport derived from ASCAT wind stress ($T'_{EK\ ASCAT}$; green), and (c) the resulting AMOC transport ($T'_{AMOC}$; black). Different colors in the upper panel indicate transport calculations for different sets of BPRs – 4 BPRs (petrol), 3 BPRs (500 m WB, 300 m EB, 500 m EB; purple) and 2 BPRs (300 m & 500 m WB; magenta) combined with the annual and semi-annual harmonics derived from the fully equipped period (05/2014-10/2015; see section 4.1).**

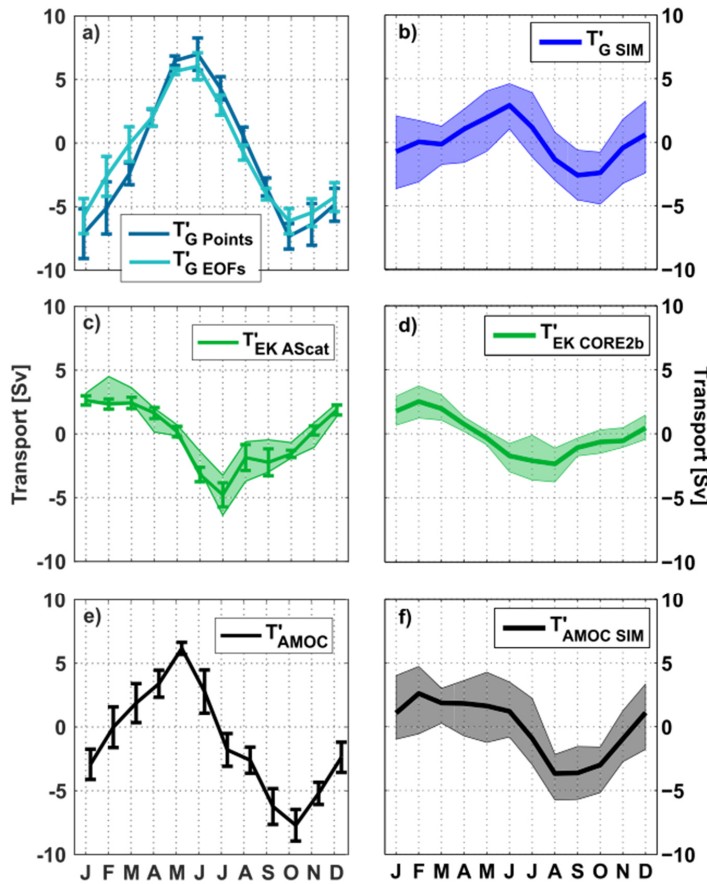


**Figure 9** Mean seasonal cycles of transport anomalies $T'_G$ (a,b), $T'_{EK}$ (c,d), $T'_{AMOC}$ (e,f). Upper-ocean geostrophic transport anomalies, $T'_{G\ Points}$ (petrol curve) and $T'_{G\ EOFs}$ (teal curve), are derived from SLA and BP observations (as described in section 4.1) and averaged over the period 2013-2018, while $T'_{G\ SIM}$, is derived from the INALT01 model velocity fields (as described in section 4.2) and averaged over the period 1978-2007. For the observations, the respective standard errors per month are given by the error bars.
$T'_{AMOC}$ in (e) was derived using $T'_{G\ EOFs}$. For the 30-year INALT01 run (b,d,f) and the 12-year ASCAT wind time series (c), shading represents the minimum and maximum range of mean seasonal cycles calculated for running 5-year windows, representing interannual variability.


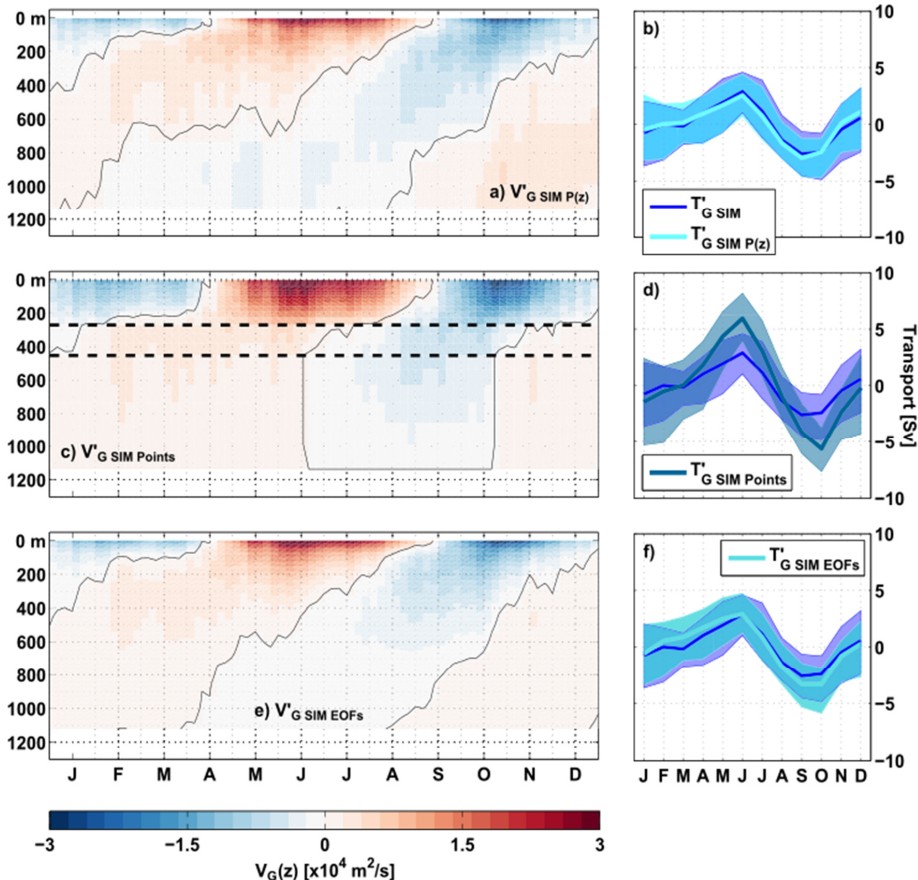

**Figure 10** Mean seasonal cycles of the geostrophic transport per unit depth, $V'_{G\,SIM}$ (a,c,e) and the upper-ocean geostrophic transport $T'_{G\,SIM}$ (b,d,f) from INALT01. $V'_{G\,SIM\,P(z)}$ (a) and $T'_{G\,SIM\,P(z)}$ (cyan curve; b), were calculated using the full vertical profiles of BP. $V'_{G\,SIM\,Points}$ (c) and $T'_{G\,SIM\,Points}$ (petrol curve; d) were reconstructed by piecewise linear interpolation of $V'_G$ between the 4 supporting points at 0, 300, 500, and 1130 m depth (black dashed lines mark the depths of the BPRs); $V'_{G\,SIM\,EOFs}$ (e) and $T'_{G\,SIM\,EOFs}$ (teak curve; f) by using the dominant vertical structure functions from INALT01. In (a,c,e) red colors show positive (northward) and blue colors negative (southward) anomalies. Blue curves in (b,d,f) represent the seasonal cycle of $T'_{G\,SIM}$ derived from the INALT01 model velocity fields (as described in section 4.2). The shading in (b,d,f) represents the minimum and maximum range of mean seasonal cycles calculated for running 5-year windows of the 30-year model run.



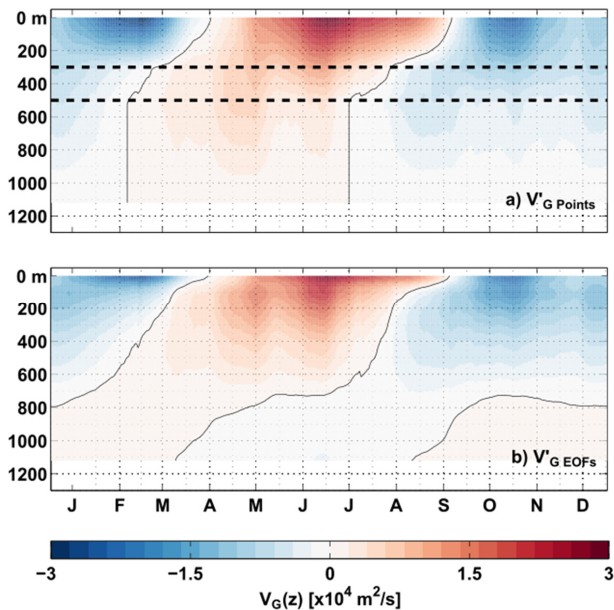

**Figure 11** Mean seasonal cycle of the geostrophic transport per unit depth, $V'_G$, over the period 2013-2018, derived from observations at 11° S with two methods: (a) Piecewise linear interpolation between the 4 supporting points at 0, 300, 500 and 1130 m depth (black dashed lines mark the depths of the BPRs). (b) Reconstruction of $V'_G$ by regression of the dominant vertical structure functions from the INALT01 model onto the values at the 3 depth levels of pressure observations at 0 m, 300 m, 500 m depth thereby relaxing the no-flow condition at 1130 m depth. Red colors show positive (northward) and blue colors negative (southward) anomalies.



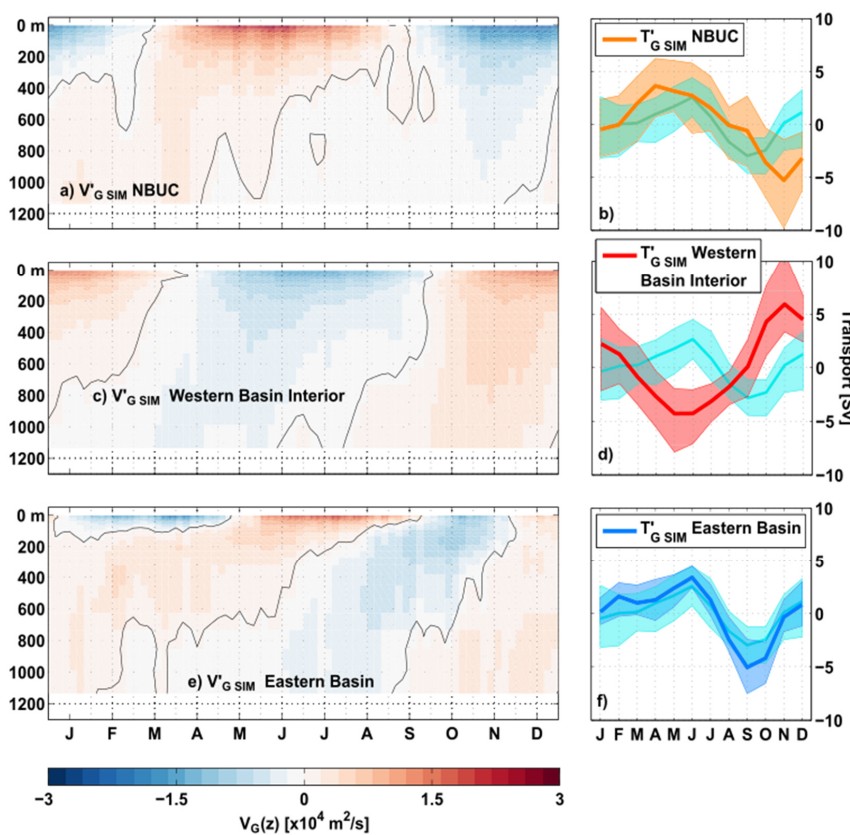

**Figure 12** Mean seasonal cycle of the geostrophic transport per unit depth, $V'_{G\,SIM}$ (a,c,e) and the upper-ocean geostrophic transport
$T'_{G\,SIM}$ (b,d,f) from INALT01. In all panels, $V'_G$ and $T'_G$ were calculated from the full vertical profiles (from the surface down to 1130
m) of the simulated pressure, but from pressure differences across different regions along 11° S: across the whole basin ($T'_{G\,SIM\,P(z)}$;
cyan curves in b, d, f); between the Brazilian continental slope and 34.55° W ($V'_{G\,SIM}\,NBUC$ in a; $T'_{G\,SIM}\,NBUC$ orange curve in b);
between 34.55° W and 15° W ($V'_{G\,SIM}\,Western\,Basin\,Interior$ in c; $T'_{G\,SIM}\,Western\,Basin\,Interior$ red curve in d); between 5°
W and the Angolan continental slope ($V'_{G\,SIM}\,Eastern\,Basin$ in e; $T'_{G\,SIM}\,Eastern\,Basin$ blue curve in f). Color coding and
shading is the same as for Fig. 10.





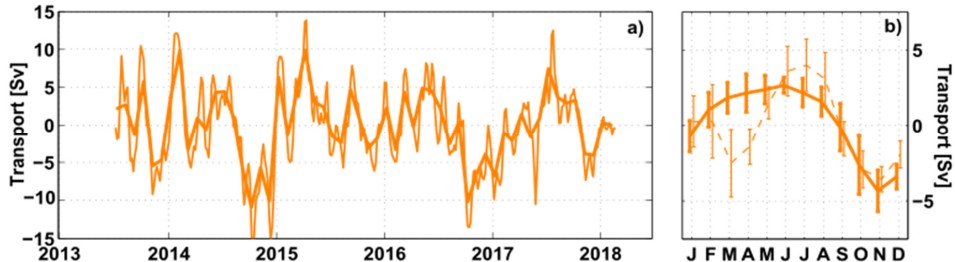

**Figure 13 a) Time series of NBUC transport anomalies (5-daily as thin curve, and monthly averages as bold curve) based on moored observations off Brazil (see section 2.4) updated from Schott et al. (2005) and Hummels et al. (2015). b) Mean seasonal cycles of the NBUC transport anomalies averaged over the periods 2013-2018 (bold curve) and 2000-2004 (thin dashed curve). The respective standard errors per month are given by the error bars.**

795

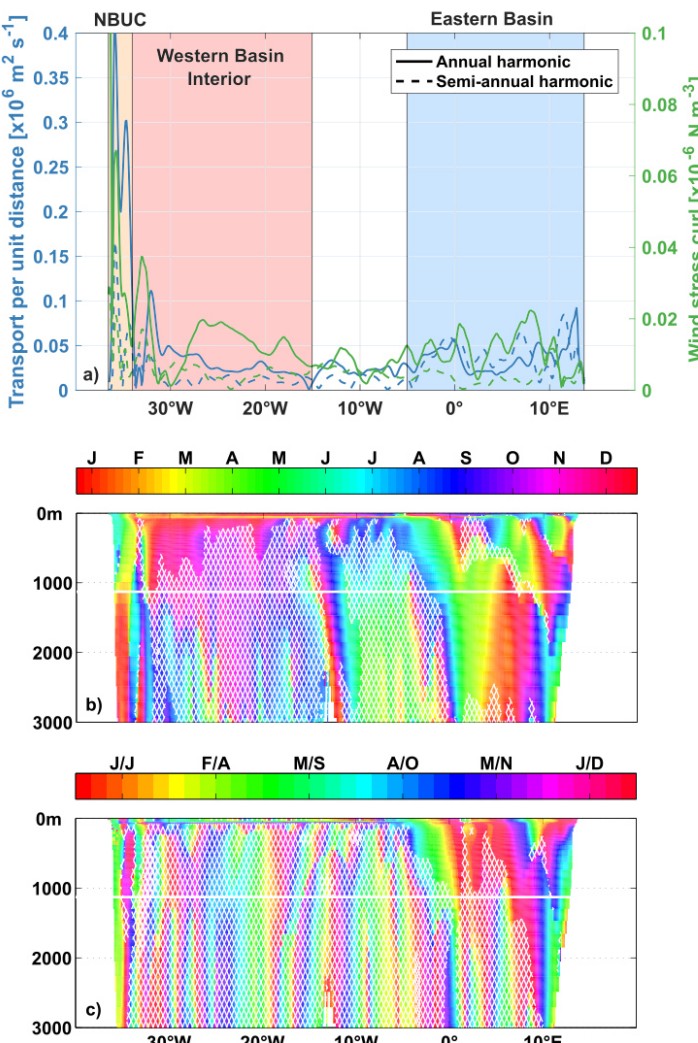

**Figure 14 a)** Amplitudes of the annual (solid curves) and semi-annual (dotted curves) harmonics of the vertically integrated upper-ocean geostrophic velocity (blue curves) and the wind stress curl (green curves) – both from INALT01 - along 11° S. Transparently shaded boxes highlight different regimes – the NBUC (orange), the western basin interior (red) and the eastern basin (blue). **b)** Zonal section of the phase of the annual harmonic of the meridional velocity in INALT01 along 11° S. White hatching overlays part of the section, where the harmonics have an amplitude < 0.0025 m/s. The white line marks the assumed level of no motion at 1130 m. **c)** Same as b), but for the semi-annual harmonic of the meridional velocity.

800