# Peer review of "Seasonal variability of the Atlantic Meridional Overturning Circulation at 11°S inferred from bottom pressure measurements"

_Ocean Science, 2020_

## Referee Comment (RC1) · Anonymous Referee #1 · 28 Jul 2020

General Comments

The study of the seasonal variability of the Atlantic Meridional Overturning Circulation (AMOC) at 11S is very important, however, some aspects of this study need clarification to warrant its publication in Ocean Science.

While any effort to extract as much information about the AMOC variability as possible from temporally and spatially sparse existent data is appreciated, the use of a model simulation (1948-2007) that does not overlap with the observations (2013-2018) is problematic. This makes it harder to pinpoint the reasons for the differences between model and observations and thus to trust the chosen observational strategy.

[Figure]

As a consequence, the seasonal cycle of AMOC transport from the observations is very different from that obtained from the model. Not only the maximum and minimum values occur in different months of the year, but also their amplitudes are statistically different.

In addition, the description of the results using different periods of time is very confusing. For instance, the periodograms of Ekman transport are presented for 2013-2018 from ASCAT dataset and for 2002-2007 from CORE2b dataset. But the minimum and maximum ranges are calculated from ASCAT for 1993-2018 and from CORE2b for 1978-2009. Even though the two datasets overlap for the period of 1993-2009, the authors then show the Hovmöller of zonal wind stress for 2008-2009. It is not clear why this is done. It would be better to compare the wind seasonal cycle obtained from both datasets for the period of 1993-2009.

Finally, the manuscript is long and most of its content is on validating the analysis rather than showing and discussing the main results about the AMOC variability. For instance, the latter is only introduced on page 11. The readability of the manuscript would also improve if information is conveyed in a more clear and straightforward way.

Specific Comments

Lines 21-22: "Here, long Rossby waves originating from equatorial forcing are known to be radiated from the Angolan continental slope and propagate westward into the basin interior." Is this shown in this study (here) or concluded from other studies? After reading the manuscript, I could not find any analysis that presents this.

Lines 103-158: Sub-sections 1.1, 1.2, 1.3 and 1.4 should be 2.1, 2.2, 2.3 and 2.4, respectively.

Line 116: I am not sure if it is necessary to describe the software used to calculate the tidal harmonics.

Lines 120-121, 133-134: Fig. 2a and Fig. 2b should be Fig. 1a and Fig. 1b, respectively.

Lines 200-205: Did the authors test other depths to have an estimate of the sensitivity of this choice (z=1130m)?

Lines 280-283 & Fig. 4: Which of these peaks are statistically significant? Particularly, considering the annual and semi-annual harmonics from 2-year long time series. This is different from the calculated uncertainty shown in shading.

Lines 297-298: Isn't this also related to the fact that the observed time series are very short and cannot capture well the annual harmonic?

Lines 309-311: Why is the periodogram for the CORE2b wind stress calculated for 2002-2007? Could a longer period from the model data be used as well to assess the impact of such observed short time series on the variability?

Lines 310-311: "The CORE2b winds do also show weak semi-annual variability, but only when considering the full time series from 1978-2009", where is this shown?

Lines 311-322: Perhaps, it would help to show an extra panel similar to panels Fig. 7c,d with the climatological evolution obtained from both dataset for 1993-2009. In fact, it is very confusing, the model outputs are for the period of 1978-2007 (Section 3). The Ekman transport periodograms are obtained from ASCAT for 2013-2018 and from CORE2b for 2002-2007. But the minimum and maximum ranges are calculated for the 1993-2018 for ASCAT and 1978-2009 for CORE2b. Why not to show a climatological Hovmöller for the overlapping period 1993-2009, instead of for 2008-2009?

Lines 336-338: The seasonal cycles of TAMOC from the observations and model are not similar. In particular, the maximum observed TAMOC occurs in May and the maximum modeled TAMOC in February, whereas the minimum observed TAMOC occurs in October and minimum modeled TAMOC in August. The amplitudes are also statistically different, comparing the error bars for the observations with the shading for the model.

Lines 359-360: This is not the case for TAMOC (previous comment).

Lines 376-389: It seems that the observation/model comparison is inconclusive.

Lines 395-403: In Fig. 12, why is the slice from 15W to 5W not included in the calculation for the interior transport? The definition of AMOC transport encompasses the whole basin, and if one wants to discuss the contributions of the WBC, interior and EBC to the AMOC variability, the slice from 15W to 5W has to be included in the interior transport. Later in lines 418-420, the authors state that there is a minimum in the annual and semi-annual harmonics in this range. However, this is not a good reason to not include the contribution from 15W-5W in the calculations. If the related transport is also minimum there, including this won't affect the main findings, but it will make the results more consistent.

Line 401-403: This is why the use of a model output that encompass the same period of the observations is so important. And also, a comparison between using shorter versus longer time series from model outputs would permit to evaluate the impact of using observed short time series on the seasonal variability.

Lines 401-427: Fig. 13a is not mentioned in the text but shows that there is not a defined seasonal cycle of the NBUC during the period of 2013-2018.

Lines 428-531: What is the impact of using the combined annual and semi-annual cycles for the eastern boundary after 11/2015 since they explain 44-61% of the variance in the daily BP time series there and for the western boundary before 05/2014 since they explain only 18-24% of the variance in this case (Lines 229-238). This was one of the main reasons to use the model outputs. Doesn't this procedure lead inevitably to the conclusion that the geostrophic transport variations are dominated by seasonal variability (Lines 466-467).

Lines 428-531: This section is too long, and the manuscript readability would benefit if most of this discussion was made in Section 5 when the authors present the results. It

is difficult to go back to figures and description of the results at this point to verify, for instance, that the structure of the meridional geostrophic velocity in the eastern basin is linked to CTW. Is this really shown in the results?

Minor Comments

Line 254: "We also test or..." should be "We also test our..."

Line 305: In "Prevailing wind stress along 11S is northwestward...", consider instead: "The prevailing winds along 11S are from southeast...".

Line 325-326: To improve readability, consider "Figure 8 displays the derived time series of TG, TEK, and the sum of both components TAMOC at 11S." instead of "Figure 8 displays the derived time series of TG, TEK, and being the sum of both components, TAMOC at 11S."

Line 412: "... to cancel out each other..." should be "... to cancel each other out..."

Line 460: "und" should be "and".

Line 752: "Hovmoeller" should be "Hovmöller"?

---

## Referee Comment (RC2) · Anonymous Referee #2 · 17 Aug 2020

This is a very nice study producing for the first time an estimate of the seasonal cycle of the meridional overturning circulation in the tropical South Atlantic along 11S using a few bottom pressure measurements (BPRs and PIES) on the boundary, satellite winds, sea level from altimetry, as well as information provided from a model (INALT01). I think that this paper reads well and the analysis presented here is important. The authors make innovative use of a few moorings to reconstruct the AMOC volume transport time series.

General comments:

1. I don't get a sense from the manuscript, how the amplitudes for AMOC seasonal

cycle transports documented at 11S compare with those at other latitudes (i.e., 26N and 34.5S) from previous studies. There is recent some evidence from observations that AMOC amplitudes decrease northward of 34.5S (i.e., Dong et al., 2015; Frajka-Williams et al. 2019; Kersale et al., 2020), and it would be nice to know how your results fit into the context of previous studies.

2. It is unclear when you report a mean +/- number whether that second number is the standard deviation, the standard error, or the uncertainty. If it is the standard error or the uncertainty, some explanation is needed for how you got to that number (i.e., how many degrees of freedom did you assume).

3. Assuming those numbers are standard deviation or standard error, that represents the variability in the time series, not the uncertainty associated with your measurement strategy. Have you made a qualitative estimate of the measurement uncertainty for each daily estimate (i.e., examined the sources of error)? If so, what is that error?

4. The figures are really nice, however, some of the figure captions are hard for me to parse. I would suggest some streamlining of the figure caption text. Some of the colors used have names that are not familiar to everyone (i.e., petrol in Fig. 8,9 and elsewhere). The colors are fine, just the nomenclature is less common (to me) and may not be familiar to all.

5. What do you think is the uncertainty in your AMOC transport associated with not having information inshore of the 300m isobath? Did you examine this within the context of the model?

6. One thing I was curious about is how much of the maximum northward volume transport (i.e., percent variance) does the Ekman vs. geostrophic volume transport account for in the observations and in the model? Do they have a very different breakdown? You talk about the amplitudes of each signal, so the result can possibly be inferred, but it is not explicitly stated in the manuscript.

7. Subsections of section 2 are labelled 1.1, 1.2, ... instead of 2.1, 2.2, ....

By line number:

21: When you say "long Rossby waves" do you mean annual Rossby waves? If so, the timescale should be mentioned at least once in abstract and in text.

33-34: "downward and upward motion... Southern Ocean" – I find this part of the sentence hard to parse, I am not sure I understood it.

48-55: A good summary paper for all of the international efforts that you may want to include if it is helpful is Frajka-Williams et al. (2019).

79: The "however" in this sentence doesn't seem needed as you are not making a contrasting statement.

85: Suggest "however they are also" instead of "but also to be"

92: Imbol Koungue et al. (2017) may also be a useful reference here, but you already have several.

99: "can even more straightforward be estimated" is a little hard to parse

120-121 and 133-134: You point to Figure 2 here but I believe you meant to point to Figure 1.

145: It is probably hard to estimate all of the uncertainties in your methodology/measurement strategy, but the errors due to winds seem possible to estimate given that you are comparing two different wind products in your study. You already do this to some extent in talking about how it affects your results.

151: Suggest "To estimate transports on the western boundary, we compute" instead of "We show"

157: Suggest "These transports are computed following methodology of Schott et al. (2005) and Hummels et al. (2015) and represent updates from their previous transport

time series" or something similar.

177: Shouldn't it be "from the western to the eastern boundary" in the parentheses?

191-192: Here a reference to other studies in the South Atlantic may be beneficial (i.e., Meinen et al. 2018; Kersale et al. 2020).

203: Are your results sensitive to your choice of 1130m as the mean depth of no motion? In the INALT01 model, how much did the depth of maximum overturning vary if you used 900m or 1300m for example?

209-213: I think you mention this in the paper, but some models don't have the right volume transport per unit depth structure (i.e., maxima is too shallow/narrow or too broad)? How well does INALT01's structure agree with the few hydrographic estimates of volume transport per unit depth that exist in the region? Maybe something to mention here or in Section 3 when model details are provided.

230: All of the other dates are month/year in the table, but here you have day/month/year. Suggest just using month/year.

233-234: I know you don't have long enough records on eastern boundary to say how robust those % variance estimates are based on 2-years of data, but you could examine whether the % variance estimates on the western boundary are sensitive to using 2- vs. 4- years of data (i.e., look at % variance in the first two years, second two years, full record)

235: Related question: Were the annual cycles from 4 years of data different from the annual cycles of 2-years of data on western boundary?

236-238: How would you estimate the uncertainty associated with only having seasonal cycle data on the eastern boundary after 11/2015? For example, if you swapped the seasonal cycle for eastern boundary time series data before 11/2015 what error do you make?

236-238: You find that the eastern boundary is more important for seasonal cycle AMOC changes, which is consistent with previous studies, but how much confidence do you have in that result given you only have 2 years of data? Confidence can be derived from the analysis of INALT01 and SLA on the eastern boundary that is shown in the paper, but perhaps this is a point to articulate more strongly.

270: Figure 4 captioning and colors are a little confusing. Please label what is SSH, pressure 300 and 500 db.

274: How do your west coast and east coast bottom pressure findings compare with Meinen et al. (2018) where they also found energy on intraseasonal and interannual time scales in ∼1000 db bottom pressure data.

282: Are the corresponding western boundary percent variances similar in the first two years as the second two years? I know the eastern boundary has more of its variance explained by those harmonics, but this would give us some sense of the stationarity of those four years.

284: "Angola was" instead of "Angola as"

285: It is unclear why there are 3 phase lines in Figure 6b given that you only have annual and semi-annual harmonic. Please clarify.

292-293: If I'm not mistaken, you aren't showing the depth dependence of the western boundary phase information (i.e., Figure 6 is only for the eastern boundary) so you could say "not shown" or point to the Figure 5 left panels.

294: You could mention here the similarities between the two 500-m deployments on the western boundary and how you get similar results. That builds more confidence in use of 2 years when you only have 2 years. (Similarly, you could break up 300-m western boundary record into two segments and compare first and second segment with full record)

297-298: Comment: It looks like the model bottom pressure seasonal cycle at 300m

and 500m on the western boundary is almost non-existent, but on the eastern boundary the model captures the pressure seasonal cycle quite well.

306: Here and elsewhere you should make clear if the +/-1.9 Sv is a standard deviation/error/uncertainty.

307: If it is standard deviation suggest replacing "an Ekman transport of" with "a mean and standard deviation of Ekman transport of." or something like that.

309: closing parentheses missing after (Fig. (7a,b)

314: In Figure 7c,d I would add years 2008 and 2009 on the left y-axis to help the reader easily follow which way time flows.

*315: I'm confused about the sign of the wind stress. Westward wind anomaly should give you southward Ekman transport anomaly (strengthening) and you say the opposite. I think the sign of the winds is wrong, not the Ekman transport that you state. This is important to sort out.

*316: Likewise, an eastward wind anomaly should give you a northward Ekman transport anomaly (weakening) and you say the opposite. I think the sign of the winds is wrong, not the Ekman transport that you state. This is important to sort out.

328: You say/show that there is good agreement during the overlapping periods, but you don't give the correlation statistics. Are the correlations high and significant?

347: "maximum northward transport in June" instead of "maximum in June"

360: at the end of this sentence please indicate the appropriate figure panel to look at (i.e., Fig. 9e,f)

390: Suggestion "the NBUC (see Section 2.4)" so that readers are reminded how you compute NBUC.

415: It is hard to see the phase propagation in Figure 14b,c – perhaps add arrows or

lines to better convey the sense of propagation.

419-420: Question: What is the depth of the mid Atlantic ridge in this region, is it deeper than 3000m?

435-436: You may want to add something here like "but clearly a longer time series will help us in the future to refine these estimates" or something like that.

442: Unclear whether "They confirm" means "Kopte et al. (2018) confirmed" or that your findings in the manuscript confirm.

480: You could compare your results to more recent studies like Meinen et al. (2018) and Kersale et al. (2020) where they look at the seasonal cycle of the MOC at 34.5S from PIES moorings which may be relevant for your study.

488: Here is one place where you can indicate if "long Rossby waves" here means "long, annual Rossby waves" (or if not annual, provide the period)

525-526: You could indicate, that long-term PIES arrays have been deployed for a decade at 34.5S in the South Atlantic (Meinen et al. 2018; Kersale et al. 2020).

Question: Some PIES moorings can be deployed with 4-year batteries and that makes it easier to determine pressure drift. Have you thought about doing so for future long-term deployments?

References:

Frajka-Williams, E., I. J. Ansorge, J. Baehr, H. L. Bryden, M. P. Chidichimo, S. A. Cunningham, G. Danabasoglu, S. Dong, K. A. Donohue, S. Elipot, P. Heimbach, N. P. Holliday, R. Hummels, L. C. Jackson, J. Karstensen, M. Lankhorst, I. A. Le Bras, M. S. Lozier, E. L. McDonagh, C. S. Meinen, H. Mercier, B. I. Moat, R. C. Perez, C. G. Piecuch, M. Rhein, M. A. Srokosz, K. E. Trenberth, S. Bacon, G. Forget, G. Goni, D. Kieke, J. Koelling, T. Lamont, G. D. McCarthy, C. Mertens, U. Send, D. A. Smeed, S. Speich, M. van den Berg, D. Volkov, and C. Wilson, 2019: Atlantic Meridional Overturning Circulation: Observed transport and variability, Frontiers in Marine Science, 6:260, doi: 10.3389/fmars.2019.00260.

Kersalé, M., C. S. Meinen, R. C. Perez, M. Le Henaff, D. Valla, T. Lamont O. T. Sato, S. Dong, T. Terre, M. van Caspel, M. P. Chidichimo, M. van den Berg, S. Speich, A. R. Piola, E. J. D. Campos, I. Ansorge, D. L. Volkov, R. Lumpkin, and S. Garzoli, 2020: Highly Variable Upper and Abyssal Overturning Cells in the South Atlantic, Science Advances, 6, eaba7573, 10.1126/sciadv.aba7573.

Imbol Koungue, R. A., S. Illig, and M. Rouault, 2017: Role of interannual Kelvin wave propagations in the equatorial Atlantic on the Angola Benguela Current system. J. Geophys. Res. Oceans, 122, 4685–4703, https://doi.org/10.1002/2016JC012463.

Meinen, C. S., S. Speich, A. R. Piola, I. Ansorge, E. D. Campos, M. Kersale, T. Terre, M. P. Chidichimo, T. Lamont, O. T. Sato, R. C. Perez, D. Valla, M. Le Henaff, S. Dong, and S. L. Garzoli, 2018: Meridional Overturning Circulation transport variability at 34.5S during 2009-2017: Baroclinic and barotropic flows and the dueling influence of the boundaries, Geophysical Research Letters, 45, 4180-4188, doi: 10.1029/2018GL077408.

---

## Author Comment (AC1) · 19 Oct 2020

**Answers to Referee #1**

*We would like to sincerely thank Referee #1 for their thoughtful comments improving our manuscript, as well as, the time and effort they put into this review. Below we address all issues raised in this review by responding to the individual comments, corrections and suggestions in italic.*

**General Comments**

**The study of the seasonal variability of the Atlantic Meridional Overturning Circulation (AMOC) at 11S is very important, however, some aspects of this study need clarification to warrant its publication in Ocean Science.**

**While any effort to extract as much information about the AMOC variability as possible from temporally and spatially sparse existent data is appreciated, the use of a model simulation (1948-2007) that does not overlap with the observations (2013-2018) is problematic. This makes it harder to pinpoint the reasons for the differences between model and observations and thus to trust the chosen observational strategy. As a consequence, the seasonal cycle of AMOC transport from the observations is very different from that obtained from the model. Not only the maximum and minimum values occur in different months of the year, but also their amplitudes are statistically different.**

*This is correct. Comparing two different periods does not allow us to analyze the correspondence of interannual variations in observations and simulations. Differences between model and observations are thus partly the result of the different periods analyzed. Unfortunately, at the moment, there is no suitable model run with atmospheric forcing covering the observed period (e.g. JRA55-do) available and sufficiently validated. However, the model is very well suited to analyze general aspects of the seasonal cycle and particularly the uncertainties of our method to derive the seasonal cycle of an AMOC time series from observations: Similar concepts are followed in Observing System Simulation Experiments (e.g. Gasparin et al., 2019; https://www.frontiersin.org/articles/10.3389/fmars.2019.00083/full).*

**In addition, the description of the results using different periods of time is very confusing. For instance, the periodograms of Ekman transport are presented for 2013-2018 from ASCAT dataset and for 2002-2007 from CORE2b dataset. But the minimum and maximum ranges are calculated from ASCAT for 1993-2018 and from CORE2b for 1978-2009. Even though the two datasets overlap for the period of 1993-2009, the authors then show the Hovmöller of zonal wind stress for 2008-2009. It is not clear why this is done. It would be better to compare the wind seasonal cycle obtained from both datasets for the period of 1993-2009.**

*We are sorry, we made a mistake here. ASCAT wind stress is available for the period 03/2007-12/2018, not 1993-2018. These numbers are corrected in the revised manuscript. When we first analyzed the model, the INALT01 model run covered the period 1978-2007. But since the CORE 2b forcing data set covers 2 more years (1978-2009), a direct comparison of ASCAT and CORE 2b wind stress in 2008-2009 (overlapping full years) is possible and performed.*

*In figure 7a, we show a periodogram of wind stress calculated for 5 years (2013-2018), for which BP measurements are available. As ASCAT covers a longer period, we can also provide an estimate for interannual variations ("minimum and maximum ranges of periodograms calculated for 5-year windows running through the full time series") plotted as an envelope. In figure 7b for CORE 2b wind stress, the 5-year period 2002-2007 was chosen arbitrarily. We understand, that this is confusing. In the revised manuscript, we do only show the envelope representing the interannual range of periodograms calculated for different 5-year subsets of the full time series.*

**Finally, the manuscript is long and most of its content is on validating the analysis rather than showing and discussing the main results about the AMOC variability. For instance, the latter is only introduced on page 11. The readability of the manuscript would also improve if information is conveyed in a more clear and straightforward way.**

*We think that the results of this study benefit from a comprehensive validation of the analysis. In order to improve the readability, we tried to go through the manuscript sentence by sentence, streamline and shorten the text as much as possible.*

**Specific Comments**

**Lines 21-22: "Here, long Rossby waves originating from equatorial forcing are known to be radiated from the Angolan continental slope and propagate westward into the basin interior." Is this shown in this study (here) or concluded from other studies? After reading the manuscript, I could not find any analysis that presents this.**

*This is concluded from other studies. In the abstract, we deleted this sentence. In other paragraphs we included Kopte et al. (2018) as the reference in which the westward propagation is discussed as part of an equatorial basin mode (see also Brandt et al., 2016):*

**Lines 103-158: Sub-sections 1.1, 1.2, 1.3 and 1.4 should be 2.1, 2.2, 2.3 and 2.4, respectively.**

*Corrected.*

**Line 116: I am not sure if it is necessary to describe the software used to calculate the tidal harmonics.**

*We would like to keep the reference, but removed the description of the software.*

**Lines 120-121, 133-134: Fig. 2a and Fig. 2b should be Fig. 1a and Fig. 1b, respectively.**

*Corrected.*

**Lines 200-205: Did the authors test other depths to have an estimate of the sensitivity of this choice (z=1130m)?**

*Yes, we did. In our study, we defined our level of no motion based on the mean depth of the zero-crossing of the meridional velocity along 11°S in the INALT01 model. And, indeed, at 11°S this depth is more variable than, for example, at 26.5°N. Most of the time (for 87% of the timesteps) this depth varies between 800m and 1300m depth, mainly following a seasonal cycle. Varying the level of no motion within this range changes the mean AMOC transport, which is largest when integrating to 1130m (14.1 Sv), by less than 10% (see Fig. 1 R1; upper panel). We added a sentence regarding this sensitivity to section 4.1. The peak-to-peak amplitude of the mean seasonal cycle of the AMOC decreases with depth – from 7.2 Sv at ~730m to 5.6 Sv at ~1470m and its minimum shifts from October to August (see Fig. 1 R1; lower panel). This is probably due to including parts of the southward lower branch when integrating to deeper levels. We added a sentence to section 4.1*

[Figure]

**Figure 1** *AMOC transport timeseries (5-daily; upper panel) and mean seasonal cycle (lower panel) at 11°S derived from the INALT01 model velocity fields over the period 1978-2007. Different colors denote different choices of a 'level of no motion' for the integration.*

**Lines 280-283 & Fig. 4: Which of these peaks are statistically significant? Particularly, considering the annual and semi-annual harmonics from 2-year long time series. This is different from the calculated uncertainty shown in shading.**

*We tested if the peaks in the periodograms of the BP time series are significant against the red noise background from an AR1 process (see Fig.2 R1; black solid curves are AR1-95%-confidence bounds and black dashed curves AR1-68%-confidence bounds). The annual cycles in the 2-year long BP time series off Angola are the only significant peaks against the 95%-confidence range of an AR1 process. When considering the 68%-confidence range then the peaks of the semi-annual cycles of Angola and at the western boundary at 500m are also statistically significant.*

[Figure]

*Figure 2* Similar to Fig.4 in the manuscript. Periodograms of BP at 300 m (upper panels) and BP at 500 m (lower panels) depth at the western (left panels) and eastern (right panels) boundaries at 11°S - calculated over the period 2013-2018. Black curves are the 95%-confidence (solid) & 68%-confidence (68%) ranges of an AR1 process.

**Lines 297-298: Isn't this also related to the fact that the observed time series are very short and cannot capture well the annual harmonic?**

*In INALT01, we are able to test how our results are affected by interannual variations and different time series lengths. Typically, we show the possible range of spectra, harmonics or mean seasonal cycles calculated for different 5-year subsets of the 30-year model run, but we also tested how the seasonal harmonics change using for example 1-year subsets, thus the total amount of available seasonal cycles of the model run and the shortest possible time series length. However, we obtain the robust result that, in INALT01, the amplitudes of the annual and semi-annual harmonics at 300m and 500m depth, and especially at the western boundary, are systematically underestimated compared to the observations.*

**Lines 309-311: Why is the periodogram for the CORE2b wind stress calculated for 2002-2007? Could a longer period from the model data be used as well to assess the impact of such observed short time series on the variability?**

*As our observational period covers about 5-years, we wanted to also show the periodogram for an arbitrary 5-year period of the CORE2b timeseries. However, the transparent envelopes, which are supposed to represent interannual variations of the results, already show the minimum and maximum ranges of periodograms calculated for 5-year windows running through the full available time series of*

*CORE2b (1978-2009) - including the subset 2002-2007. From the comments we understand, that this was rather confusing than helpful. We deleted the solid curve in Fig. 7b and tried to modify the related text and figure caption in the revised version accordingly.*

**Lines 310-311: "The CORE2b winds do also show weak semi-annual variability, but only when considering the full time series from 1978-2009", where is this shown?**

*In Fig. 7b, the transparent envelope representing interannual variations shows a second, smaller peak at a frequency of 180 days. As this peak is not statistically significant we deleted this sentence in the revised manuscript.*

**Lines 311-322: Perhaps, it would help to show an extra panel similar to panels Fig. 7c,d with the climatological evolution obtained from both dataset for 1993-2009. In fact, it is very confusing, the model outputs are for the period of 1978-2007 (Section 3). The Ekman transport periodograms are obtained from ASCAT for 2013-2018 and from CORE2b for 2002-2007. But the minimum and maximum ranges are calculated for the 1993-2018 for ASCAT and 1978-2009 for CORE2b. Why not to show a climatological Hovmöller for the overlapping period 1993-2009, instead of for 2008-2009?**

*We are sorry for the confusion, which, we think, is mainly caused by a mistake we made in the caption of Fig. 7. ASCAT wind stress is available for the period 03/2007-12/2018, not 1993-2018. The INALT01 model run covers the period 1978-2007. But since the CORE 2b forcing data set covers 2 more years (1978-2009), a direct comparison of ASCAT and CORE 2b wind stress in 2008-2009 (overlapping full years) was possible and performed.*

**Lines 336-338: The seasonal cycles of TAMOC from the observations and model are not similar. In particular, the maximum observed TAMOC occurs in May and the maximum modeled TAMOC in February, whereas the minimum observed TAMOC occurs in October and minimum modeled TAMOC in August. The amplitudes are also statistically different, comparing the error bars for the observations with the shading for the model.**

*This is correct. In the revised manuscript, in addition to the minimum and maximum ranges of mean seasonal cycles calculated for running 5-year windows running through the respective available periods we do also show the total range of possible values per month. Even when considering the total range, the mean seasonal cycles of $T'_G$ and $T'_{AMOC}$ are just outside the total range of possible results in INALT01. We rephrased several related sentences and tried to be more specific on where we find good agreement and where we find differences. However, we think that the model is very well suited to analyze the relevant mechanisms and test our method to derive the seasonal cycle of an AMOC time series from observations.*

**Lines 359-360: This is not the case for TAMOC (previous comment).**

*Please see the previous comment.*

**Lines 376-389: It seems that the observation/model comparison is inconclusive.**

*We now more clearly state where we find good agreement and where we find differences between model simulations and observations (see above).*

**Lines 395-403: In Fig. 12, why is the slice from 15W to 5W not included in the calculation for the interior transport? The definition of AMOC transport encompasses the whole basin, and if one wants to discuss the contributions of the WBC, interior and EBC to the AMOC variability, the slice from 15W to 5W has to be included in the interior transport. Later in lines 418-420, the authors state that there is a minimum in the annual and semi-annual harmonics in this range. However, this is not a good reason to not include the contribution from 15W-5W in the calculations. If the related transport is**

**also minimum there, including this won't affect the main findings, but it will make the results more consistent.**

*This is absolutely right. We extended the slices of the Western basin Interior and eastern basin to 10°W, which corresponds approximately with the location of the Mid-Atlantic Ridge crest. The results change very little. The minimum of the seasonal cycle of the eastern basin contribution, however, is reduced by half its amplitude and shifted from September to August. A small maximum in February is gone. Extending the eastern basin slice to 10°W results in an even greater similarity between its seasonal cycle and the seasonal cycle of the basin-wide upper-ocean geostrophic transport.*

**Line 401-403: This is why the use of a model output that encompass the same period of the observations is so important. And also, a comparison between using shorter versus longer time series from model outputs would permit to evaluate the impact of using observed short time series on the seasonal variability.**

*As stated above, at the moment, there is no suitable model run with atmospheric forcing covering the observed period (e.g. JRA55-do) available and sufficiently validated. We hope for this to happen in the near future. In order to give an estimate for possible interannual variations, all mean seasonal cycles are shown together with the total range of possible values (single years) and, for the 30-year model run, the range calculated for 5-year subsets. We tried to highlight this more clearly in the text.*

**Lines 401-427: Fig. 13a is not mentioned in the text but shows that there is not a defined seasonal cycle of the NBUC during the period of 2013-2018.**

*Fig 13a is now mentioned in section 2.4. Despite the large year-to-year variability in the seasonal cycle, a mean seasonal cycle can be obtained (shown in Fig. 13b) that can be compared to a mean seasonal cycle and its variability as obtained from the model simulations.*

**Lines 428-531: What is the impact of using the combined annual and semi-annual cycles for the eastern boundary after 11/2015 since they explain 44-61% of the variance in the daily BP time series there and for the western boundary before 05/2014 since they explain only 18-24% of the variance in this case (Lines 229-238). This was one of the main reasons to use the model outputs. Doesn't this procedure lead inevitably to the conclusion that the geostrophic transport variations are dominated by seasonal variability (Lines 466-467).**

*We tested replacing certain BPRs with the corresponding combined annual and semi-annual harmonics in the fully equipped period 05/2014 - 11/2015 (cf. Fig.8a). In those 18 months, the correlation between the daily $T'_G$ time series derived with 4 BPRs ("$T'_{G\ EOFs\ 4\ BPRs}$") and the daily $T'_G$ time series with the WB 300m BPR replaced ("$T'_{G\ EOFs\ 3\ BPRs}$") is high R=0.97 and the correlation with $T'_G$ derived with 2 WB BPRs and the EB combined annual and semi-annual harmonics ("$T'_{G\ EOFs\ 2\ BPRs}$") is R=0.85. As the latter can explain ~70% of the total variance in $T'_G$ over the fully equipped period, we are still confident to capture most of the variability in $T'_G$ after 11/2015. As this period is only 18 months, we assume the 30% of the variance that we seem to miss with our method to be related to the intra-seasonal signals we see in the spectra for EB BP (Fig.4 d,f). However, it is correct, that with our methods we give too much weight to seasonal variability compared to other timescales, but on the other hand we believe it is the best we can make out of the currently available time series.*

*Having measurements only at 2 depths and the surface, the main reason to use the model, was to understand and approximate the vertical structure of $V'_G$ (z). We do not use the model to fill data gaps or replace missing sensors.*

**Lines 428-531: This section is too long, and the manuscript readability would benefit if most of this discussion was made in Section 5 when the authors present the results. It is difficult to go back to**

**figures and description of the results at this point to verify, for instance, that the structure of the meridional geostrophic velocity in the eastern basin is linked to CTW. Is this really shown in the results?**

*We tried to shorten this section as much as possible and shifted parts of the discussion to section 5. In the manuscript, we also tried to clarify, that only the vertical structure and variability of the pressure at the eastern boundary can be related to CTWs – not the vertical structure of the meridional geostrophic velocity integrated over the whole eastern basin.*

**Minor Comments**

**Line 254: "We also test or: : :" should be "We also test our: : :"**

*Corrected. Thanks.*

**Line 305: In "Prevailing wind stress along 11S is northwestward: : :", consider instead: "The prevailing winds along 11S are from southeast: : :".**

*We changed the sentences accordingly.*

**Line 325-326: To improve readability, consider "Figure 8 displays the derived time series of TG, TEK, and the sum of both components TAMOC at 11S." instead of "Figure 8 displays the derived time series of TG, TEK, and being the sum of both components, TAMOC at 11S."**

*Changed.*

**Line 412: ": : : to cancel out each other: : :" should be ": : : to cancel each other out: : :"**

*Corrected.*

**Line 460: "und" should be "and".**

*Corrected.*

**Line 752: "Hovmoeller" should be "Hovmöller"?**

*Corrected.*

**References**

*Brandt, P., Claus, M., Greatbatch, R. J., Kopte, R., Toole, J. M., Johns, W. E., and Böning, C. W.: Annual and semiannual cycle of equatorial Atlantic circulation associated with basinmode resonance. J. Phys. Oceanogr., 46, 3011–3029, https://doi.org/10.1175/JPO-D-15-0248.1, 2016.*

*Gasparin, F., Guinehut, S., Mao, C., Mirouze, I., Rémy, E., King, RR., Hamon, M., Reid, R., Storto, A., Le Traon, P-Y., Martin, M.J. and Masina, S.: Requirements for an integrated in situ Atlantic Ocean observing system from coordinated observing system simulation experiments. Front. Mar. Sci., 6, 83, https://doi.org/10.3389/fmars.2019.00083, 2019.*

---

## Author Comment (AC2) · 19 Oct 2020

**Answers to Referee #2**

**This is a very nice study producing for the first time an estimate of the seasonal cycle of the meridional overturning circulation in the tropical South Atlantic along 11S using a few bottom pressure measurements (BPRs and PIES) on the boundary, satellite winds, sea level from altimetry, as well as information provided from a model (INALT01). I think that this paper reads well and the analysis presented here is important. The authors make innovative use of a few moorings to reconstruct the AMOC volume transport time series.**

*We would like to sincerely thank this referee for their kind words, the time and effort they put into this review and the helpful suggestions improving our manuscript. Below we address the issues raised in this review by responding to the individual comments from Referee #2 in italic.*

General comments:

**1. I don't get a sense from the manuscript, how the amplitudes for AMOC seasonal cycle transports documented at 11S compare with those at other latitudes (i.e., 26N and 34.5S) from previous studies. There is recent some evidence from observations that AMOC amplitudes decrease northward of 34.5S (i.e., Dong et al., 2015; Frajka- Williams et al. 2019; Kersale et al., 2020), and it would be nice to know how your results fit into the context of previous studies.**

*As seasonal AMOC variability is closely related to variations in the regional wind regimes, we do not expect similar amplitudes or phases of the seasonal cycle of the AMOC in the Tropics compared to the Subtropics or Subpolar regions. Therefore, comparing our results directly to the seasonal cycle of the AMOC observed at RAPID or SAMBA does not seem to be extremely useful with regard to the local mechanisms, but indicate the importance of understanding the seasonal cycle when trying to extract longer-term variability from observations. Thus, we added a list of the seasonal cycle amplitudes of the AMOC at RAPID and SAMBA to the discussion: "In the Subtropics, recent estimates of the peak-to-peak amplitude of the mean seasonal cycle of the AMOC range from 4.3 Sv at 26.5°N (2004-2017; Frajka-Williams et al., 2019) to 13 Sv at 34.5°S (2014-2017; Kersale et al., 2020)." Thank you for pointing out the study by Kersale et al. (2020) – we had not been aware of it. The seasonal cycle of the AMOC at 11°S is similar in amplitude compared to SAMBA and much stronger than at RAPID.*

**2. It is unclear when you report a mean +/- number whether that second number is the standard deviation, the standard error, or the uncertainty. If it is the standard error or the uncertainty, some explanation is needed for how you got to that number (i.e., how many degrees of freedom did you assume).**

*In the submitted manuscript, all error bars plotted together with mean seasonal cycles (fig.9 (a,c,e) and fig.13 b) were standard errors $\left( = \frac{\sigma}{\sqrt{N}} \right)$ with σ being the standard deviations and N the number of available years. In the revised version of the manuscript – as a response to other comments - we show the absolute range of possible values as dashed curves instead.*

*In the text, the "±"-numbers provided together with estimates of mean transports are also standard errors $\left( = \frac{\sigma}{\sqrt{M_{eff}}} \right)$, but with $M_{eff} = \frac{M}{n_d}$, where M is defined as the length of the time series and $n_d$ as the decorrelation time scale. To clarify this, we added a sentence to the methods ("In the following, all mean transports are presented together with the standard error $SE = \sigma \Big/ \sqrt{N/n_d}$, where σ is the standard deviation and nd the decorrelation time scale of the respective time series of length N.") and changed a sentence in section 5.3 to "This is within the uncertainty range of 3 Sv for the AMOC estimate of 16.2 Sv derived from a hydrographic ship section along 11° S in 1994 (Lumpkin & Speer, 2007)."*

**3. Assuming those numbers are standard deviation or standard error, that represents the variability in the time series, not the uncertainty associated with your measurement strategy. Have you made a qualitative estimate of the measurement uncertainty for each daily estimate (i.e., examined the sources of error)? If so, what is that error?**

*Many problems associated with the measurements of the bottom pressure recorders, like the sensor drift or de-tiding, are thoroughly discussed in Kanzow et al. (2006). We followed their procedure and tried to document all the steps and choices we made during the processing of the BPRs. These are the uncertainties associated with the instruments themselves.*

*The second part of uncertainties arises from the observational strategy: With BP observations only at 2 depths and SLA at the surface, we had to test the uncertainties associated with this observing system and our methods to approximate the vertical structure of the basin-wide geostrophic transport above 1000m. To do this we use the INALT01 model. INALT01 was found to produce comparable variations in the Western Boundary Current transport at 11°S (Hummels et al., 2015) and, therefore, considered to be a good choice for this analysis. Of course, the model is not perfect. However, even if there are differences between the model and the reality, we can use the model "reality" to test our observational strategy meaning that with the help of the model, we were able to quantify uncertainties of the derived transports that are related to different aspects of the observing system.*

*That's why large parts of our manuscript are already very technical, as pointed out by Reviewer #1. We consider the effect of the uncertainty of each daily estimate on the seasonal variability of geostrophic transports small compared to the uncertainty introduced by the different aspects of the observing system, mainly because of the dominance of the annual and semi-annual variability compared to variability on other time scales.*

**4. The figures are really nice, however, some of the figure captions are hard for me to parse. I would suggest some streamlining of the figure caption text. Some of the colors used have names that are not familiar to everyone (i.e., petrol in Fig. 8,9 and elsewhere). The colors are fine, just the nomenclature is less common (to me) and may not be familiar to all.**

*We tried to streamline all of the figure captions as well as to simplify the color scheme and nomenclature.*

**5. What do you think is the uncertainty in your AMOC transport associated with not having information inshore of the 300m isobath? Did you examine this within the context of the model?**

*We tested this in the INALT01 model. There, the transport in the western boundary wedge inshore of the 300m isobath is -0.04 ± 0.002 Sv, hence negligible. And from velocity ship sections available along 11°S, we estimate even smaller transports between 0.007 -0.008 Sv for the 300m western boundary wedge.*

**6. One thing I was curious about is how much of the maximum northward volume transport (i.e., percent variance) does the Ekman vs. geostrophic volume transport account for in the observations and in the model? Do they have a very different breakdown? You talk about the amplitudes of each signal, so the result can possibly be inferred, but it is not explicitly stated in the manuscript.**

*Considering 5-year subsets of the 30 years of the INALT01 model run, the seasonal variability (repeated mean seasonal cycles) of $T_{EK\ CORE2b}$ can explain between 10-19% and $T_{G\ SIM}$ between 20-28% of the total variance in $T_{AMOC\ SIM}$. In the observations, which cover the years 2013-2018, seasonal variability of $T'_{EK\ ASCAT}$ and $T'_{G}$ (using 5-day averages as in INALT01) can explain 13% and 48% of the total variance in $T'_{AMOC}$, respectively. This last number is, of course, strongly dependent on our method. Using the*

*combined annual and semi-annual harmonics whenever a sensor is missing results in over-weighting of seasonal variability while missing variability on other timescales.*

**7. Subsections of section 2 are labelled 1.1, 1.2, ... instead of 2.1, 2.2, : : :.**

*Corrected.*

**By line number:**

**21: When you say "long Rossby waves" do you mean annual Rossby waves? If so, the timescale should be mentioned at least once in abstract and in text.**

*This sentence was rephrased, but we modified this formulation whenever used elsewhere.*

**33-34: "downward and upward motion: : : Southern Ocean" – I find this part of the sentence hard to parse, I am not sure I understood it.**

*Changed to "……through water mass transformation, for example, in the subpolar North Atlantic or near the Southern Ocean."*

**48-55: A good summary paper for all of the international efforts that you may want to include if it is helpful is Frajka-Williams et al. (2019).**

*Indeed, it is a really good summary paper. We added this reference.*

**79: The "however" in this sentence doesn't seem needed as you are not making a contrasting statement.**

*Deleted.*

**85: Suggest "however they are also" instead of "but also to be"**

*Changed as suggested.*

**92: Imbol Koungue et al. (2017) may also be a useful reference here, but you already have several.**

*Thanks for the suggestion. We added the reference.*

**99: "can even more straightforward be estimated" is a little hard to parse**

*Changed to "But, circulation changes in z-coordinates can also be estimated by measuring the pressure differences between the eastern and western boundary at each depth."*

**120-121 and 133-134: You point to Figure 2 here but I believe you meant to point to Figure 1.**

*Corrected.*

**145: It is probably hard to estimate all of the uncertainties in your methodology/ measurement strategy, but the errors due to winds seem possible to estimate given that you are comparing two different wind products in your study. You already do this to some extent in talking about how it affects your results.**

[Figure]

[Figure]

*Figure 1 Ekman transport time series (5-daily; upper panel) and mean seasonal cycles (lower panel) derived from ASCAT (blue) and CORE2b (red) wind stress for the overlapping years 2008-2009.*

*In the manuscript we list the mean Ekman transports and standard errors for the two different products, we compare their respective periodograms and mean seasonal cycles together with estimate of interannual variations beyond the observational period. Additionally, Fig. 1 R2 (upper panel) shows the Ekman transport time series derived from 5-daily averages of ASCAT and 5-daily CORE2b wind stress for the overlapping years 2008-2009, which are highly correlated (R=0.82; upper panel). The mean seasonal cycles (Fig.1 R2; lower panel) calculated for the overlapping two years, show differences in amplitude and shifts up to 1 month between extrema. In comparison to the mean seasonal cycle we estimated for the upper-ocean geostrophic contribution to the AMOC seasonal cycle (cf. Fig. 9a in the manuscript), the differences between these two wind products – as estimated for 2008-2009 - are just within the uncertainty of the two methods we use to approximate the vertical structure of the geostrophic transport. We think that further investigations would be beyond the scope of this article.*

**151: Suggest "To estimate transports on the western boundary, we compute" instead of "We show"**

*Changed to "…To estimate the western boundary current transport, we compute….".*

**157: Suggest "These transports are computed following methodology of Schott et al. (2005) and Hummels et al. (2015) and represent updates from their previous transport time series" or something similar.**

*We re-arranged the paragraph, but would like to keep the information as the methodology is slightly different in Schott et al. (2005) and Hummels et al. (2015).*

**177: Shouldn't it be "from the western to the eastern boundary" in the parentheses?**

*Yes, correct. Thanks.*

**191-192: Here a reference to other studies in the South Atlantic may be beneficial (i.e., Meinen et al. 2018; Kersale et al. 2020).**

*As the methods between this study (using only the BPRs and SLA) are different the methods used to estimate the AMOC at SAMBA (travel time from PIES for "baroclinic" and BP for "barotropic" components), we think, adding these references here does not support the statement. However, we added Kersale et al. (2020) to the introduction when the efforts at SAMBA are introduced.*

[Figure]

**203: Are your results sensitive to your choice of 1130m as the mean depth of no motion? In the INALT01 model, how much did the depth of maximum overturning vary if you used 900m or 1300m for example?**

*Yes, they are, but within the range of the other methodological uncertainties. Indeed, at 11°S this depth of no motion is more variable than, for example, at 26.5°N. Most of the time (for 87% of the timesteps) it varies between 800m and 1300m depth, mainly following a seasonal cycle. Varying the level of no motion within this range (see Fig.2 R2; upper panel) changes the mean AMOC transport, which is largest when integrating to 1130m (14.1 Sv), by less than 10%. The peak-to-peak amplitude of the mean seasonal cycle (Fig.2 R2; lower panel) of the AMOC decreases with depth – from 7.2 Sv at ~730m to 5.6 Sv at ~1470m and its minimum shifts from October to August.*

*Figure 2 AMOC transport timeseries (5-daily; upper panel) and mean seasonal cycle (lower panel) at 11°S derived from the INALT01 model velocity fields over the period 1978-2007. Different colors denote different choices of a 'level of no motion' for the integration.*

[Figure]

**209-213: I think you mention this in the paper, but some models don't have the right volume transport per unit depth structure (i.e., maxima is too shallow/narrow or too broad)? How well does INALT01's structure agree with the few hydrographic estimates of volume transport per unit depth that exist in the region? Maybe something to mention here or in Section 3 when model details are provided.**

*The only study we are aware of that presents an estimate of the overturning streamfunction at 11°S based on observations is by Lumpkin & Speer (2003, see Fig. 3 R2; right panel). They calculated their overturning streamfunction from the WOCE A8 section along 11°S conducted in 1994 and in defined neutral density classes. The results from INALT01 (Fig.3 R2; left panel) show good agreement regarding the vertical structure and ampitude of the overturning stream function when considering the range of possible variations around the mean over the 30-years of the model run. We added this information to section 3.*

*Figure 3 Overturning streamfunction across 11°S with neutral density classes on the y-axis. Left panel: Derived from the INALT01 model velocity field over the period 1978-2007. Black curves give the one and two standard deviations around the 30-year mean. Right panel: Derived from the hydrographic WOCE sections A8 in 1994, copied directly from Lumpkin & Speer, 2003 (their Fig.8). The shading indicates standard error bars.*

**230: All of the other dates are month/year in the table, but here you have day/month/year. Suggest just using month/year.**

*Thanks. Corrected.*

**233-234: I know you don't have long enough records on eastern boundary to say how robust those%variance estimates are based on 2-years of data, but you could examine whether the % variance estimates on the western boundary are sensitive to using 2- vs. 4- years of data (i.e., look at % variance in the first two years, second two years, full record)**

*Thanks for the suggestion. We did: The table below lists the % of variances explained by the annual (AC), semi-annual (SAC) and combined harmonics fitted to 2-year subsets of the WB BPR timeseries.*

*We find the combined annual and semi-annual harmonics to always explain a similar fraction of the overall variance in the time series. But, the fractions related to the annual or semi-annual harmonic changes between the first and second halves of both BP time series (300m & 500m) at the western*

*boundary. It seems that during the period 2014-2015 the semi-annual cycle was dominant and during the period 2016-2017 the annual cycle. This behavior can also be seen in figure 1. While this shift from one dominant time scale to the other is interesting, it is far beyond the scope of this study.*

| | Explained variance [%] | | | | | |
|---|---|---|---|---|---|---|
| | 300m WB (KPO 1134) | | | 500m WB (KPO 1135) | | |
| | AC | SAC | A&SAC | AC | SAC | A&SAC |
| 2014-2015 | 9,1 | 18,0 | 27,4 | 6,1 | 24,5 | 31,6 |
| 2016-2017 | 22,2 | 2,8 | 26,3 | 20,4 | 8,0 | 30,4 |
| full | 17,0 | 6,7 | 24,1 | 14,6 | 12,8 | 28,5 |

**235: Related question: Were the annual cycles from 4 years of data different from the annual cycles of 2-years of data on western boundary?**

[Figure]

**Figure 4** *Annual (left panels) and semi-annual (right panels) harmonics calculated for 2-year subsets - solid for 2014-2015 and dashed for 2016-2017 - of the available western boundary BP time series (similar to Fig. 5c,e in the manuscript) at 300m (upper panels) and 500m (lower panels).*

*Shown in Fig.4 R2 are the annual and semi-annual harmonics calculated for 2-year subsets of the available western boundary BP time. Both, the annual and semi-annual harmonics show pronounced differences in their amplitudes (up to 50%) and minor differences in their phases (<1 month) between the two 2-year periods. Interestingly, the ratio of the amplitudes of the annual or semi-annual harmonic change between the two periods – please see the previous comment. However, the combined annual and semi-annual harmonics explain a similar fraction of the overall variance in both subsets of and in the full time series.*

**236-238: How would you estimate the uncertainty associated with only having seasonal cycle data on the eastern boundary after 11/2015? For example, if you swapped the seasonal cycle for eastern boundary time series data before 11/2015 what error do you make?**

*Within the fully equipped period 05/2014 - 11/2015 the correlation between the daily $T'_G$ time series derived with 4 BPRs and the daily $T'_G$ time series derived with WB BPRs and the EB combined annual and semi-annual harmonics is high (R=0.85) and statistically significant. The latter can explain ~70% of the total variance in $T'_G$ over fully equipped period. As this period is only 18 months, we assume the 30% of the variance that we seem to miss with our method to be related to the intra-seasonal signals we see in the spectra for EB BP (Fig.4 d,f).*

**236-238: You find that the eastern boundary is more important for seasonal cycle AMOC changes, which is consistent with previous studies, but how much confidence do you have in that result given you only have 2 years of data? Confidence can be derived from the analysis of INALT01 and SLA on the eastern boundary that is shown in the paper, but perhaps this is a point to articulate more strongly.**

*We added this sentence to the manuscript: "We derive confidence in our method from the comparison of the observed BP variations with variations in the simulated BP time series and in the SLA time series off Angola, both covering longer periods."*

**270: Figure 4 captioning and colors are a little confusing. Please label what is SSH, pressure 300 and 500 db.**

*We added this information to the y-axis labels in Fig.4 & 5.*

**274: How do your west coast and east coast bottom pressure findings compare with Meinen et al. (2018) where they also found energy on intraseasonal and interannual time scales in _1000 db bottom pressure data.**

*In contrast to the findings at 34.5°S, at 11°S, the BP time series at the eastern boundary exhibit more energy on intra-seasonal to seasonal time scales. Although, 90d and 120d peaks are found at 34.5°S as well as at 11°S, any comparison of intraseasonal variability between those two latitudes is difficult considering the differences in local wind regimes, impact or non-impact of equatorial forcing and eddy activity. Interannual variation of the SLA at the eastern boundary are, as discussed in the manuscript, thought to be related to equatorial dynamics and should not compare to findings at 34.5°S. We do also find hints for more energy on interannual timescales at the western boundary, but the time series are just not long enough for reliable estimates.*

**282: Are the corresponding western boundary percent variances similar in the first two years as the second two years? I know the eastern boundary has more of its variance explained by those harmonics, but this would give us some sense of the stationarity of those four years.**

*See our answer to some previous comments.*

**284: "Angola was" instead of "Angola as"**

*Corrected.*

**285: It is unclear why there are 3 phase lines in Figure 6b given that you only have annual and semi-annual harmonic. Please clarify.**

*We show phase lines for the minimum of the annual harmonic in black and for the two minima of the semi-annual harmonic during a year in grey. The description was changed to "…phases of the minima of the annual and semi-annual harmonics…".*

**292-293: If I'm not mistaken, you aren't showing the depth dependence of the western boundary phase information (i.e., Figure 6 is only for the eastern boundary) so you could say "not shown" or point to the Figure 5 left panels.**

*That's correct. As can be seen in Fig.5, amplitudes of the seasonal harmonics are small at the western boundary and phases very uncertain. As suggested, we added references to the corresponding panels in Figure 5 to the text.*

**294: You could mention here the similarities between the two 500-m deployments on the western boundary and how you get similar results. That builds more confidence in use of 2 years when you only have 2 years. (Similarly, you could break up 300-m western boundary record into two segments and compare first and second segment with full record)**

*We added a sentence to section 5.1: "This is also consistent between 2-year subsets of the western boundary BP time series."*

**297-298: Comment: It looks like the model bottom pressure seasonal cycle at 300m and 500m on the western boundary is almost non-existent, but on the eastern boundary the model captures the pressure seasonal cycle quite well.**

*Exactly. We rephrased this sentenced to "The model tends to overestimate the annual harmonic at the surface and generally underestimate seasonal variability in general at depth - especially at the western boundary the seasonal cycle of the simulated BP at 300 m and 500m depth is almost non-existent."*

**306: Here and elsewhere you should make clear if the +/-1.9 Sv is a standard deviation/ error/uncertainty.**

*Done.*

**307: If it is standard deviation suggest replacing "an Ekman transport of" with "a mean and standard deviation of Ekman transport of." or something like that.**

*Rephrased.*

**309: closing parentheses missing after (Fig. (7a,b)**

*Added, thanks.*

**314: In Figure 7c,d I would add years 2008 and 2009 on the left y-axis to help the reader easily follow which way time flows.**

*Done as suggested.*

**315: I'm confused about the sign of the wind stress. Westward wind anomaly should give you southward Ekman transport anomaly (strengthening) and you say the opposite. I think the sign of the winds is wrong, not the Ekman transport that you state. This is important to sort out.**

*You are right, the sign of the wind stress anomalies was wrong. Thanks for pointing this out. We corrected it.*

**316: Likewise, an eastward wind anomaly should give you a northward Ekman transport anomaly (weakening) and you say the opposite. I think the sign of the winds is wrong, not the Ekman transport that you state. This is important to sort out.**

*See previous comment.*

**328: You say/show that there is good agreement during the overlapping periods, but you don't give the correlation statistics. Are the correlations high and significant?**

*Within the period 05/2014-10/2015, when 5 BPRs were in place, we found high and statistically significant (p<0.001) correlations of 0.97 and 0.85 between the geostrophic transport time series estimated from the full set of 4 BPRs and estimated from 3 BPRs (WB 300m missing and replaced with combined annual & semi-annual harmonics) or 2 BPRs (EB 300m & 500m missing and replaced with combined annual & semi-annual harmonics), respectively.*

**347: "maximum northward transport in June" instead of "maximum in June"**

*Changed as suggested.*

**360: at the end of this sentence please indicate the appropriate figure panel to look at (i.e., Fig. 9e,f)**

*Done.*

**390: Suggestion "the NBUC (see Section 2.4)" so that readers are reminded how you compute NBUC.**

*It does not make sense to us to include the cross-reference to section 2.4 in this paragraph. We added it to the next paragraph: "Having a mooring array installed off the coast off Brazil measuring the Western Boundary Current system there (e.g. Hummels et al., 2015; see section 2.4), allows us to directly compare the seasonal variability of the NBUC in INALT01 with observations.".*

**415: It is hard to see the phase propagation in Figure 14b,c – perhaps add arrows or lines to better convey the sense of propagation.**

*Figures 14b,c were removed in the process of "streamlining" our manuscript.*

**419-420: Question: What is the depth of the mid Atlantic ridge in this region, is it deeper than 3000m?**

*Along 11°S, the top of the MAR is at about 2700m water depth. Actually, in the first submitted version of the manuscript it could be seen as a small white peak in figures 14 (b,c).*

**435-436: You may want to add something here like "but clearly a longer time series will help us in the future to refine these estimates" or something like that.**

*Added as suggested.*

**442: Unclear whether "They confirm" means "Kopte et al. (2018) confirmed" or that your findings in the manuscript confirm.**

*We changed the sentence to "Kopte et al. (2018) confirm....."*

**480: You could compare your results to more recent studies like Meinen et al. (2018) and Kersale et al. (2020) where they look at the seasonal cycle of the MOC at 34.5S from PIES moorings which may be relevant for your study.**

*Interestingly, the seasonal cycle of the geostrophic transport derive from PIES in Meinen et al. (2018) is very different to the estimate derived from ARGO/WOA presented in Dong et al. (2014). However, we deleted this discussion in order to streamline and shorten the manuscript. As we show in our study, seasonal variations in the geostrophic AMOC contribution at 11° S are driven by processes in the tropical Atlantic region. Therefore, a direct comparison between the Tropics and Subtropics on these timescales does not seem to be useful with regard to the mechanisms at work.*

**488: Here is one place where you can indicate if "long Rossby waves" here means "long, annual Rossby waves" (or if not annual, provide the period)**

*Changed to "annual".*

**525-526: You could indicate, that long-term PIES arrays have been deployed for a decade at 34.5S in the South Atlantic (Meinen et al. 2018; Kersale et al. 2020).**

*At SAMBA, the strategy to estimate the AMOC and its variability relies not only on BP measurement, but also on the acoustic travel times measured by the PIES on both sides of the basin (e.g. Meinen et al., 2017). The same approach was also used before to estimate the North Atlantic Current transport at 47° N (NOAC array; Roessler et al., 2015). This is different to our strategy mainly based on BP measurements. While at the WB we have PIES installed at 300m and 500m depth, the BP time series off Angola are measured with single BPRs.*

*Here, we wanted to make the point, that there is also a lot of potential in using only the BP measurements. However, it would be interesting to test how the travel times derived from the PIES installed off Brazil can add information to or reduce the uncertainty of our results. We hope to perform the respective analyses in the future.*

**Question: Some PIES moorings can be deployed with 4-year batteries and that makes it easier to determine pressure drift. Have you thought about doing so for future longterm deployments?**

*Thank you for the suggestion. We think, the best option would be to have more temporal overlapping observational periods of PIES, which however, requires more instruments. Nevertheless, we try to have the longest possible deployment periods.*

**References:**

**Frajka-Williams, E., I. J. Ansorge, J. Baehr, H. L. Bryden, M. P. Chidichimo, S. A. Cunningham, G. Danabasoglu, S. Dong, K. A. Donohue, S. Elipot, P. Heimbach, N. P. Holliday, R. Hummels, L. C. Jackson, J. Karstensen, M. Lankhorst, I. A. Le Bras, M. S. Lozier, E. L. McDonagh, C. S. Meinen, H. Mercier, B. I. Moat, R. C. Perez, C. G. Piecuch, M. Rhein, M. A. Srokosz, K. E. Trenberth, S. Bacon, G. Forget, G. Goni, D. Kieke, J. Koelling, T. Lamont, G. D. McCarthy, C. Mertens, U. Send, D. A. Smeed, S. Speich, M. van den Berg, D. Volkov, and C. Wilson, 2019: Atlantic Meridional Overturn- ing Circulation: Observed transport and variability, Frontiers in Marine Science, 6:260, doi: 10.3389/fmars.2019.00260.**

**Kersalé, M., C. S. Meinen, R. C. Perez, M. Le Henaff, D. Valla, T. Lamont O. T. Sato, S. Dong, T. Terre, M. van Caspel, M. P. Chidichimo, M. van den Berg, S. Speich, A. R. Piola, E. J. D. Campos, I. Ansorge, D. L. Volkov, R. Lumpkin, and S. Garzoli, 2020: Highly Variable Upper and Abyssal Overturning Cells in the South Atlantic, Science Advances, 6, eaba7573, 10.1126/sciadv.aba7573.**

**Imbol Koungue, R. A., S. Illig, and M. Rouault, 2017: Role of interannual Kelvin wave propagations in the equatorial Atlantic on the Angola Benguela Current system. J. Geophys. Res. Oceans, 122, 4685–4703, https://doi.org/10.1002/2016JC012463.**

**Meinen, C. S., S. Speich, A. R. Piola, I. Ansorge, E. D. Campos, M. Kersale, T. Terre, M. P. Chidichimo, T. Lamont, O. T. Sato, R. C. Perez, D. Valla, M. Le Henaff, S. Dong, and S. L. Garzoli, 2018: Meridional Overturning Circulation transport variability at 34.5S during 2009-2017: Baroclinic and barotropic flows and the dueling influence of the boundaries, Geophysical Research Letters, 45, 4180-4188, doi: 10.1029/2018GL077408.**

*Dong, S., Baringer, M. O., Goni, G. J., Meinen, C. S., and Garzoli, S. L.: Seasonal variations in the South Atlantic Meridional Overturning Circulation from observations and numerical models, Geophys. Res. Lett., 41, 4611– 4618, doi:10.1002/2014GL060428, 2014.*

*Hummels, R., Brandt, P., Dengler, M., Fischer, J., Araujo, M., Veleda, D., and Durgadoo, J. V.: Interannual to decadal changes in the western boundary circulation in the Atlantic at 11° S, Geophys. Res. Lett., 42, doi:10.1002/2015GL065254, 2015.*

*Kanzow, T., Cunningham, S. A., Johns, W. E., Hirschi, J. J., Marotzke, J., Baringer, M. O., Meinen, C. S., Chidichimo, M. P., Atkinson, C., Beal, L. M., Bryden, H. L., and Collins, J.: Seasonal Variability of the Atlantic Meridional Overturning Circulation at 26.5° N. J. Climate, 23, 5678–5698, doi:10.1175/2010JCLI3389, 2010.*

*Meinen, C. S., Garzoli, S. L., Perez, R. C., Campos, E., Piola, A. R., Chidichimo, M.-P., Dong, S., and Sato, O. T.: Characteristics and causes of Deep Western Boundary Current transport variability at 34.5° S during 2009-2014, Ocean Sci., 175-194, doi:10.5194/os-13-175-2017, 2017.*

*Lumpkin, R., and K. Speer, 2003: Large-Scale Vertical and Horizontal Circulation in the North Atlantic Ocean. J. Phys. Oceanogr., 33, 1902–1920, https://doi.org/10.1175/1520-0485(2003)033<1902:LVAHCI>2.0.CO;2.*

*Roessler, A., Rhein, M., Kieke, D., and Mertens, C.: Long-term observations of North Atlantic Current transport at the gateway between western and eastern Atlantic. J. Geophys. Res.-Oceans, 120,4003–4027. doi:10.1002/2014JC010662, 2015.*

*Schott, F. A., Dengler, M., Zantopp, R., Stramma, L., Fischer, J., and Brandt, P.: The Shallow and Deep Western Boundary Circulation of the South Atlantic at 5° −11° S. J. Phys. Oceanogr., doi:10.1175/JPO2813.1, 2005.*